# Biotic Response of Plankton Communities to Middle to Late Miocene Monsoon Wind and Nutrient Flux Changes in the Oman Margin Upwelling Zone

Gerald Auer[1], Or M. Bialik[2,3], Mary-Elizabeth Antoulas[1], Noam Vogt-Vincent[4], Werner E. Piller[1]

[1]University of Graz, Department of Earth Sciences, NAWI Graz Geocenter, Heinrichstrasse 26, 8010 Graz, Austria
[2]University of Muenster, Institute of Geology and Palaeontology, Corrensstr. 24, 48149 Münster, Germany
[3]Dr. Moses Strauss Department of Marine Geosciences, The Leon H. Charney School of Marine Sciences, University of Haifa, Carmel 31905, Israel.
[4]Department of Earth Sciences, University of Oxford, Oxford, UK

*Correspondence to: Gerald Auer (gerald.auer@uni-graz.at) & Or M. Bialik (obialik@uni-muenster.de)*

**Keywords**

Indian summer monsoon, upwelling, Miocene, calcareous nannoplankton, intermediate waters, nutrient fluxes

**Abstract.** Understanding past dynamics of upwelling cells is an important aspect of assessing potential upwelling changes in future climate change scenarios. Our present understanding of nutrient fluxes throughout the world's oceans emphasizes the importance of intermediate waters transporting nutrients from the Antarctic divergence into the middle and lower latitudes. These nutrient-rich waters fuel productivity within wind-driven upwelling cells in all major oceans. One such upwelling system is located along the Oman Margin in the Western Arabian Sea (WAS). Driven by cross-hemispheral winds, the WAS upwelling zone's intense productivity led to the formation of one of the most extensive oxygen minimum zones known today.

In this study covering the Middle to Late Miocene at ODP Site 722, we investigate the inception of upwelling-derived primary productivity. This study presents new plankton assemblage data in the context of existing model- and data-based evidence constraining the tectonic and atmospheric boundary conditions for upwelling in the WAS. With this research, we build upon the original planktonic foraminifer-based research by Dick Kroon in 1991 as part of his research based on the Ocean Drilling Project (ODP) LEG 117.

We show that monsoonal winds likely sustained upwelling since the emergence of the Arabian Peninsula after the Miocene Climatic Optimum (MCO) ~14.7 Ma, with fully monsoonal conditions occurring since the end of the Middle Miocene Climatic Transition (MMCT) ~13 Ma. However, changing nutrient fluxes through Antarctic Intermediate and sub-Antarctic Mode Waters (AAIW/SAMW) were only established after ~12 Ma. Rare occurrences of diatoms frustules correspond to the maximum abundances of *Reticulofenestra haqii* and *Reticulofenestra antarctica*, indicating higher upwelling-derived nutrient levels. By 11 Ma, diatom abundance increases significantly, leading to alternating diatom blooms and high-nutrient-adapted nannoplankton taxa. These changes in primary producers are also well reflected in geochemical proxies with increasing $\delta^{15}N_{org.}$ values (> 6‰) and high organic carbon accumulation. These proxies provide further independent evidence for high productivity and the onset of denitrification simultaneously.

Our multi-proxy-based evaluation of Site 722 primary producers provides evidence for a stepwise evolution of Middle to Late Miocene productivity in the western Arabian Sea for the first time. The absence of a clear correlation with existing deep marine climate records suggests that local processes, such as monsoonal wind conditions but crucially also hints at changing lateral nutrient transport through upwelling intermediate waters, likely played an important role in modulating productivity in the western Arabian Sea. Finally, we show that using a multi-proxy record provides novel insights into how plankton responded to changing nutrient conditions through time in a monsoon-wind-driven upwelling zone.

## 1. Introduction

Within coastal upwelling zones, wind-driven Ekman transport brings nutrient-rich deep water into the photic zone (Woodward et al., 1999). This process supports enhanced primary productivity in the surface ocean. This increased productivity supports a large biomass across the entire food chain, reaching far afield from the core of the upwelling zone. In addition, the high productivity in upwelling zones produces a significant amount of marine snow (both organic and inorganic), which sinks through the water column. As the organic particulates fall, they become partially remineralized, consuming oxygen and forming an oxygen-depleted zone. (Morrison et al., 1998; McCreary et al., 2013)However, the flux of organic matter is so large that a significant volume of organic matter reaches and accumulates on the seafloor (e.g., Suess, 1980; Rixen et al., 2019a, b).

Upwelling zones affect the marine carbon cycle by sequestering carbon and exchanging carbon between the ocean and the atmosphere via the dissolved inorganic carbon system and $p$CO$_2$ changes (Rixen et al., 2006; Krapivin and Varotsos, 2016; Wang et al., 2015). Increased photosynthesis-driven primary productivity during upwelling produces high organic carbon export from the photic zone into the deep sea through the organic carbon pump (Volk and Hoffert, 1985; Ridgwell and Zeebe, 2005). Primary producers account for most of the biomass in upwelling zones, with phytoplankton accounting for > 80% of the particulate organic carbon (Head et al., 1996). Calcification by these primary producers and heterotrophic organisms feeding on them is further an important contributor to the in-organic carbon cycle of the oceans (Falkowski, 1997; Raven and Falkowski, 1999; Ridgwell and Zeebe, 2005; Millero, 2007). However, the productivity of coastal upwelling zones highly depends on atmospheric conditions as they are primarily wind-driven. Consequently, wind-driven upwelling further constitutes a direct intersection between the oceans and the atmosphere. Hence, changes in average wind speeds are directly responsible for the intensity and size of upwelling zones (Dugdale, 1972; Shimmield, 1992; Tudhope et al., 1996; Balun et al., 2010). Therefore, these atmospheric processes may also influence the community structure of primary producers and consumers within the area affected by upwelling.

(Lee et al., 1998; Honjo et al., 1999; Munz et al., 2017; Rixen et al., 2019b)oday, the Western Arabian Sea (WAS) upwelling is one of the most productive marine regions (Lee et al., 1998; Honjo et al., 1999; Munz et al., 2017; Rixen et al., 2019b). Its high productivity and organic matter flux fuels the Arabian Sea oxygen minimum zone (OMZ), which extends southwards from the Oman Margin between 200 and 1000 m water depth, reaching as far south as 10°N (Morrison et al., 1998; McCreary et al., 2013), making it one of the largest oxygen deficient zones in the modern ocean.

Primary productivity in the WAS is furthermore driven by seasonal winds flowing norward along the east coast of Africa (Currie et al., 1973; Rixen et al., 2019a) as an extension of the Somali/Findlater Jets (Sarr et al., 2022; Findlater, 1969). Upwelling in the WAS is thus directly forced by the cross-hemispheric circulation system of the Indian Summer Monsoon (Findlater, 1969; Woodward et al., 1999; Basavani, 2013; Sarr et al., 2022). The prevailing southwesterly winds in the region during the summer months result in the displacement of large water masses (Tudhope et al., 1996; Schott and McCreary, 2001; Schott et al., 2009; Lahiri and Vissa, 2022), resulting in pronounced, intense upwelling peaks during the summer monsoon season (Lee et al., 1998; Honjo et al., 1999; Rixen et al., 2019b). During the northern hemisphere winter, the prevailing wind direction in the Arabian Sea reverses as a weaker and dryer winter monsoon becomes established (Gadgil, 2018). The northeasterly winter monsoon winds result in an additional, albeit less pronounced, productivity spike in the region (Madhupratap et al., 1996; Munz et al., 2015, 2017; Rixen et al., 2019b). Between these two regimes – the inter-monsoon season – weak and variable winds dominate, permitting the establishment of well-stratified regions in the WAS that exhibit oligotrophic surface water conditions. The shift between the different conditions generates a complex pattern of abundance shifts between nutrient-adapted and primarily meso- but potentially even oligotrophic phytoplankton communities. This dynamic impact of changes in wind regimes and upwelling intensity on plankton communities in the WAS is well-established for the modern (Schiebel et al., 2004).

In the Arabian Sea, significant variability in productivity has been identified over Pleistocene glacial-interglacials. For example, higher productivity in the Late Pleistocene is associated with interglacial periods (Schubert et al., 1998; Pourmand et al., 2007; Avinash et al., 2015; Naik et al., 2017). Conversely, these climatically driven changes in primary productivity affect the volume of the oxygen minimum zone (OMZ) and the intensity of denitrification in the region (Gaye et al., 2018). An OMZ is the result of the complete consumption of dissolved in the water

column due to the microbial degradation of sinking organic matter. Hence OMZ strength is generally related to the strength of primary productivity and, thus, organic matter flux within the overlying upwelling cell (Dickens and Owen, 1994; McCreary et al., 2013; Stramma et al., 2008)

Based on current records, the earliest activity within the upwelling zone may have occurred earlier in the Burdigalian (Bialik et al., 2020b). However, it was not until connectivity to the proto-Mediterranean was terminated, and the Arabian Peninsula began to emerge that the regional geographic configuration allowed the establishment of a strong upwelling cell driven by the Findlater Jets (Rögl, 1999; Reuter et al., 2013; Harzhauser et al., 2007; Bialik et al., 2019; Sarr et al., 2022). After the Miocene Climatic Optimum (MCO) ~14 Ma (Flower and Kennett, 1994; Frigola et al., 2018; Sosdian and Lear, 2020), global cooling resumed, and a stable, upwelling zone and a sustained OMZ resembling present-day conditions were reported to have established in the WAS (Kroon et al., 1991; Zhuang et al., 2017; Bialik et al., 2020a).

Modelling studies suggest that the inception of upwelling and the WAS was closely linked to the tectonic evolution of the Arabian Peninsula, which resulted in water displacement by the Findlater Jet along a newly emergent coastline of Oman (Zhang et al., 2014; Sarr et al., 2022). Therefore, the uplift of the Arabian Peninsula is now seen as the dominant controlling factor for the inception of monsoonal upwelling in the WAS, which is now also seen as largely separate from prevailing monsoonal rainfall patterns (Sarr et al., 2022). After the tectonic configuration of the Arabian Peninsula was in place, the cross-hemispheric wind patterns of the South Asian Monsoon were subsequently able to drive upwelling in the WAS in a near modern configuration since the MMCT (Bialik et al., 2020a; Betzler et al., 2016; Gupta et al., 2015).

Evidence suggests that strong upwelling in the Arabian Sea first occurred between the Middle and Late Miocene (Kroon et al., 1991; Huang et al., 2007a; Tripathi et al., 2017; Zhuang et al., 2017; Bialik et al., 2020a; Alam et al., 2022). To date, manganese redirection – i.e., the depletion of Mn in the sedimentary record due to Mn-reduction in the water column and subsequent advective transport to the edges of the OMZ – is one of the most used proxies to define OMZs and their past extent within the ocean (Dickens and Owen, 1994). Together with sedimentological facies and micropaleontological studies (Dickens and Owen, 1999; Gupta et al., 2004) these methods have been used effectively to track the size of the OMZ throughout the Indian Ocean and, by proxy, also the intensity of upwelling derived primary productivity. $\delta^{15}N$ values > 6 ‰ are seen as possible indicators for significant water column denitrification within the OMZ based on the approach of Tripathi et al. (2017). Bialik et al. (2020a) applied this approach for a Middle to Late Miocene interval at Site 722, showing that upwelling in the WAS may have sustained an OMZ strong enough for denitrification to occur as early as 11 Ma ago. However, these methods do not provide direct evidence for how changing wind and nutrient levels have interacted to result in the observed OMZ pattern.

Following these lines of evidence, it can be summarized that WAS upwelling initiated during the Middle to Late Miocene during the Middle Miocene Climatic Transition (MMCT), marked by cooling sea surface temperatures (SSTs) since ~14.7 Ma (Zhuang et al., 2017; Holbourn et al., 2014, 2015). Monsoonal winds subsequently intensified only after the MMCT at ~13 Ma, in conjunction with OMZ expansion to the Maldives (Betzler et al., 2016) before reaching maximum intensity at ~11 Ma and potentially declining at ~9 Ma (Bialik et al., 2020a). Upwelling re-intensified later in the Miocene and oscillated into the Plio-Pleistocene (Kroon et al., 1991; Huang et al., 2007b; Gupta et al., 2015; Tripathi et al., 2017; Alam et al., 2022). The Serravallian upwelling intensification is accompanied by significantly increased biogenic silica accumulation across the northern Indian Ocean (Keller and Barron, 1983; Baldauf et al., 1992). This biogenic silica bloom is dominated by siliceous plankton such as

diatoms and radiolaria (Nigrini, 1991), indicating a sustained regime of high nutrient levels, which was able to support these primary producers (Blain et al., 1997; Schiebel et al., 2004; Mikaelyan et al., 2015).

The present study aims to better constrain the relationships and interactions between different plankton groups in the WAS within the context of the dynamic changes occurring in the Oman Margin upwelling cell throughout the Middle to Late Miocene.

## 2.     ODP Site 722 – Site location, age model, and oceanographic setting

Ocean Drilling Project (ODP) Site 722 (16°37'18.7" N/59°47'45.33" E) lies offshore Oman on the Owen Ridge, a 300-km-long and 50-km wide feature in the WAS (Fig. 1a). Site 722 is located at a water depth of 2027.8 m (Shipboard-Scientific-Party, 1989) at the edge of the present-day Oman upwelling zone (Fig. 1a), and lies below the core of the Indian Ocean Oxygen Minimum Zone (OMZ), with oxygen concentrations $< 2$ µmol kg$^{-1}$ persisting at a depth between c. 200 – 1000 m water depth (McCreary et al., 2013; Garcia et al., 2018).

The sedimentary cover at the site location comprises nannofossil, foraminifer, and diatom-rich pelagic oozes, with silty clay (Shipboard-Scientific-Party, 1989; Rodriguez et al., 2014; Bialik et al., 2020a). Bialik et al. (2020a) recently published a revised age model for Site 722, which we will utilize throughout this study. The age-depth correlation relies on biostratigraphic information from the nannofossil assemblage data used in this study, combined with existing shipboard data (Shipboard-Scientific-Party, 1989). The age model covers the study interval over the Middle Miocene to the Late Miocene (c. 15.0 – 8.5 Ma, corresponding to a core depth of 276.62 to 404.94 mbsf). Bialik et al. (2020a) published benchtop x-ray fluorescence (XRF)-based elemental data, total organic carbon content (TOC), and the calcite equivalent carbonate content in the analyzed samples. These geochemical proxy data were subsequently used in conjunction with the nannofossil assemblage data to fully constrain the response of the assemblage to changing environmental conditions in the WAS upwelling zone.

The modern-day water mass configuration of the WAS (Fig. 1b) indicates that Indian Central Water (ICW) upwells in the upwelling region offshore oman. The ICW result from a mixture of warm, highly saline Red Sea and Persian Gulf Waters (RSPGW), as well as Sub-Antarctic Mode and Intermediate Waters (SAWM, and AAIW respectively). Modern oceanographic research suggests that AAIW/SAMW, which contributes to the ICW is the dominant source of nutrients in the Arabian Sea upwelling region today (Böning and Bard, 2009; Toggweiler et al., 2019a; You and Tomczak, 1993; You, 1997, 1998). In addition, at present, there also exists some contribution of the Indonesian Intermediate Waters (IIW), the ICW in the WAS (Fig. 1a and 1b). Therefore, changes in the supply of these water masses are a critical aspect of understanding the region's past and likely future upwelling dynamics (Böning and Bard, 2009; Laufkötter and Gruber, 2018; Toggweiler et al., 2019b). The Middle to Late Miocene was similar to the modern (Bialik et al., 2019; Hall, 2012). However, the Indonesian Throughflow region's configuration remains largely enigmatic, with potentially large emergent island chains and extensive coral reefs between Australia and South East Asia (Hall, 2012). Deep and Intermediate water exchange and, thus, IIW formation may thus have been restricted in the Miocene. If present, IIW likely would supply additional nutrients, including a significant amount of bioavailable silica, to the upwelling zone in the WAS (You and Tomczak, 1993; You, 1997). Waters in the WAS therefore repesent mixture of SAMW/AAIW and IIW with ICW, which later intermix with the regionally formed RSPGW (Böning and Bard, 2009; Toggweiler et al., 2019a).

## 3. Methods

### 3.1. Nannofossil and siliceous fragment quantification

We produced smear slides from 71 freeze-dried samples taken from Hole 722B (supplementary data 1) following the quantitative drop technique of Bordiga et al. (2015). On each slide, at least 47 field views were counted until at least 300 specimens were recorded or until over 190 field views were reached for samples containing very low abundances. During counting, nannofossils were identified down to the species level whenever possible. The occurrence of diatom frustules (including pennate and centric forms), as well as other biogenic silica fragments (including silicoflagellates and radiolarian fragments), were quantitatively recorded without further taxonomic identification (supplementary data 1). All recorded nannofossil taxa (+ siliceous fragments) were then converted into absolute abundances per g/sediment, according to Bordiga et al. (2015), with portions of the dataset already published (Bialik et al., 2020a). In addition to the above-described quantification, the high amount of biogenic silica recorded in some sections often dilutes absolute nannofossil abundances, to alleviate the issues with potential dilution of nannofossil abundance due to high fluxes of biogenic silica, we calculated nannofossil and siliceous fragment fluxes for the studied interval (see section 3.5).

#### Taxonomic Remarks

3.1.1. We relied on the Nannotax3 website (Nannotax 3, 2023) for detailed taxonomic reference and identification. In addition, taxonomic identification followed the concepts outlined in Perch-Nielsen (1985) and Young (1998), the Handbook of Calcareous Nannoplankton 1–5 (Aubry, 1984, 1988, 1989, 1990, 1999), and the compilation on the taxonomy of the order Discoasterales by Aubry (2021).

For subsequent ecological interpretations, we combined the identified *Reticulofenestra* morphotypes into three size bins ranging from small (<3 μm) to medium (<3-5 μm) and large (>5 μm). There is some debate regarding the taxonomic distinction of the reticulofenestrids (genus *Reticulofenestra*) in the Neogene (see Young, 1998, for discussion). Several research groups (Auer et al., 2019; Gibbs et al., 2005; Imai et al., 2017; Jatiningrum and Sato, 2017; Wade and Bown, 2006) apply different size ranges to differentiate *Reticulofenestra* taxa based on placolith size. We also note that each of these size ranges may contain a multitude of genotypes (Young, 1998). In this study, we follow the species concept of Auer et al. (2019) adapted for the Middle to Late Miocene:

- *Reticulofenestra* spp. (small) cf. *R. minuta*: reticulofenestrids < 3 μm in length without a bar spanning the central area.
- *Reticulofenestra haqii*: reticulofenestrids 3–5 μm in length with an open central area.
- *Reticulofenestra antarctica*: reticulofenestrids 3–5 μm in length with a closed central area.
- *Reticulofenestra pseudoumbilicus* (small): all reticulofenestrids 5–7 μm in length.
- *Reticulofenestra pseudoumbilicus* (sensu stricto): all reticulofenestrids >7 μm in length.

### 3.2. Planktonic foraminifera counts and quantification

For foraminifera analysis, 28 samples were freeze-dried, weighed, and wet-sieved using mesh sizes 250, 125, and 63 μm. After sieving, sample residues were oven dried at 40°C. For quantitative foraminiferal analyses, the size fractions > 250 μm and 250-125 μm were examined under a stereomicroscope (Zeiss V8). In each sample, at least 200 specimens were picked and identified. In 8 samples, less than 200 specimens were found in the available material. When necessary, samples were split into smaller aliquots (splits). The total number of foraminifera in the

sediment (N/g) was calculated from the number of the counted specimen and the number of splits. Relative
abundances (%) were calculated for each species (see supplementary data 2 for details).

## 3.3.    Statistical Analyses and Ordination

All applied statistical and ordination methods were performed using PAST4 (v. 4.11 released 2022-09-13; Hammer
et al., 2001). The applied methods include correlation matrices between nannofossil taxa and XRF-based
environmental proxy data for dust flux and Mn depletion, the abundance of siliceous fragments, and calcite
equivalent $CaCO_3$ content (supplementary data 3). Percentage data were then arcsine-transformed before cluster
analyses and ordination methods. The arcsine transformation was applied to generate a statistically viable dataset
suitable for the applied clustering and ordination methods (Sokal and Rohlf, 1995; Hammer and Harper, 2006;
Auer et al., 2014, 2019; Bialik et al., 2021) and applies the universal paired group method with arithmetic mean
(UPGMA) with Bray-Curtis distance. Cluster stability was further evaluated by using UPGMA clustering with
Euclidian distance and Ward's method.
The contributing taxa of each cluster were subsequently evaluated based on similarity percentage (SIMPER)
analysis (Bray-Curtis similarity). The correspondence of nannofossil variability within each sample with
environmental parameters was investigated using the non-metric multidimensional scaling (nMDS), where
geochemical proxy data (see sect. 2; Fig. 3) were used as environmental variables and visualized as vectors within
the two-dimensional coordinate space of the nMDS. Additionally, several diversity indices (see supplementary
data 1), including the Shannon H'-diversity, were automatically calculated for the calcareous nannofossil
assemblage (Hammer and Harper, 2006).

## 3.4.    Published geochemical proxy data used in this study

In addition to the paleobiological data generated for this study, we further apply a suite of previously published
geochemical proxy data (Bialik et al., 2020a), which we utilize as additional lines of evidence to anchor the
observed assemblage variation within a multiproxy framework. In brief, we apply CaCO3 and TOC combined
with fluxes of siliceous fragments (see section 3.5 for details), as productivity proxies. Benchtop x-ray
fluorescence-derived elemental ratios further supplement this interpretation, where we apply Mn/Al ratios to
quantify Mn redirection (see Bialik et al., 2020a), based on the model of Dickens and Owen (1994). The available
XRF data was also used to generate a dust flux proxy based on the elemental ratio of (K+Al)/(Fe+Ti+Zr), as
defined by Kuhnt et al. (2015). This dust flux proxy allows determining the accumulation of Fe, Ti and Zr bearing
heavy mineral phases, compared to elements predominantly present in clay minerals (Al + K). We interpret this
proxy as a qualitative proxy for wind-derived dust flux and, thus, varying wind strength at Site 722. Dustflux and
wind speed are intrinsically linked to Africa's progressive aridification due to the uplift of the Arabian Peninsula
(Zhang et al., 2014; Sarr et al., 2022). The published $\delta^{15}N$ is also discussed in the context of the new assemblage
data. Tripathi et al. (2017) interpret $\delta^{15}N$ values > 6 ‰ as an indicator for significant water column denitrification
in ocean basins with oxygenated bottom waters. Later,  Bialik et al. (2020a) also used this proxy interpretation for
the Middle to Late Miocene interval at Site 722, which will be followed herein.
**3.5.    Calculation of accumulation rates and fluxes**
To quantifiy flux rates we applied moisture and density (MAD) derived bulk density data generated during Leg
117 (Shipboard-Scientific-Party, 1989), to calculate mass accumulation rates (MAR). To calculate bulk MARs we
applied linear interpolated dry bulk density for each sample point using the calculation
$$BMAR = \frac{DBD \ x \ LSR}{10}$$

Where BMAR ist the bulk mass accumulation rate in g/cm2/kyr, and DBD is the dry bulk density in g/cm$^3$ based
on shipboard MAD data, and LSR is the linear sedimentation rate in m/myr calculated based on the age model of
Bialik et al. (2020a). Thusly generated bulk MARs where subsequently used to also calkulate mass fluxes of TOC,
CaCO$_3$ given as g/cm$^2$/Ma. Fossil fluxes are given as nannofossil accumulation rates (NAR) as well as diatom
accumulation rates (DAR), which are calculated by multiplying the BMAR with the number of individuals per g
of sediment.
**4.    Results**
**4.1.    Calcareous Nannofossils**
**Nannofossil abundance, diversity**
[4.1.1.]Nannofossil preservation found to be good to moderately good based on visual evaluation using light and scanning
electron microscopy. Overall preservation in biogenic-silica-rich samples was noted to be slightly poorer than in
samples with little or no biogenic silica.
Total nannofossil fluxes range from 4,77*8 to 9.93*10$^{10}$ liths/cm$^2$/Ma, with an average of 1.45*10$^{10}$ and a median
of 1.07*10$^{10}$. By comparison, total nannofossils per g/sed. range from 2.75*10$^8$ to 4.11*10$^{10}$ with an average of
5.73*10$^9$ and a median of 4.04*10$^9$. Diatom accumulation range from no frustules to 2.41*10$^{10}$ frustules/cm$^2$/kyr,
with an average of 2.24*10$^9$ and a median of 3.72*10$^8$. In the three uppermost samples taken from Core 722B-
30X, small placolith abundance (primarily *Reticulofenestra minuta*) increases sharply above the base absence (Ba)
of *Reticulofenestra pseudoumbilicus* (Backman et al., 2012; Agnini et al., 2017) after 8.8 Ma (Fig. 2). For details
[4.1.2.]on the abundance and variability of individual nannofossil taxa, please refer to the supplementary material
(supplementary data 1).
**Clusters and Ordination**
Cluster analysis (UPGMA, Bray-Curtis similarity) resulted in 4 major clusters (clusters 1-4) that were defined at
a similarity cutoff of 0.61 with a cophenetic correlation coefficient of 0.81. Clusters 1 and 4 were again split into
2 (clusters 1a-b) and 3 (clusters 4a-c) sub-clusters, respectively, at a similarity cutoff of 0.66 (Fig. 4a).
Bootstrapping (N=1000) shows weak support for individual clusters reflecting the overall strong similarities in the
assemblage composition of the studied samples. However, one-way ANOSIM shows p-values of <0.05, indicating
that the separated clusters are statistically significant.
Based on SIMPER analysis, the clusters and subclusters are primarily defined by the abundance variability of
reticulofenestrids, discoasterids, *Cyclicargolithus floridanus,* and, to a smaller extent, *Coccolithus pelagicus,* and
*Sphenolithus* spp. Based on these results, we infer that the clusters represent taphogroups, each reflecting different
environmental conditions (see Auer et al., 2014).
Taphogroup (TG) 1a is characterized by a very high abundance of small reticulofenestrids. TG 1b is similarly
characterized by a high abundance of small reticulofenestrids, although lower than TG 1a, with a higher abundance
of medium reticulofenestrids and *Cyclicargolithus floridanus*. TG 2 is characterized by a high abundance of *C.
floridanus,* and TG 3 by a high abundance of large reticulofenestrids with common discoasterids. TG 4 and its
subgroups are primarily defined by the variation of the three size ranges of reticulofenestrids, with TG4a exhibiting
the highest abundances of small reticulofenestrids, TG4b showing the lowest amounts of both small and medium
reticulofenestrids, and through TG4c high numbers of both medium and large reticulofenestrids. See table 1 for a
summary of the TGs and the supplementary material (supplementary data4) for a statistical breakdown of the
contribution of all taxonomic groups to each TG.
The cluster analysis results are well represented within the nMDS, with TGs splitting well along coordinates 1 and
2. Furthermore, the recorded stress of the nMDS is 0.13, indicating that the results are robust (Clarke, 1993). We,
however, note the overall high compositional similarity of clusters, particularly sub-clusters, which results in
higher stress in the nMDS. This is important, as recently a more conservative approach has been put forward,
recommending that nMDS outputs exhibiting stress above 0.1 should be carefully evaluated (Bialik et al., 2021).
We found a positive loading for TOC, and siliceous fragments, along coordinates one and two. Dustflux, calculated
as $\ln((Zr+Ti+Fe)/(Al+K))$ following Kunt et al. (2015), is positively loaded on coordinate one but negatively
loaded on coordinate two. The Mn/Al ratio is loaded negatively on coordinate 1 and positively on coordinate 2.
Whereas $CaCO_3$ is loaded negatively on both coordinates (Fig. 4b).
**4.2.    Planktonic Foraminifera**
Out of 28 samples one sample (722B-34X-3W 30-32, ca. 10.2 Ma) was barren in planktonic foraminifera. In the
remaining 27 samples, 27 taxa of planktonic foraminifera were identified. The planktonic foraminifera
perservation was overall good, but decreases downhole. The foraminifera tests were found to be moderately
pyritized. Of these taxa, 5 (*Globigerinoides ruber*, *Globorotalia menardii*, *Neogloboquadrina acostaensis*,
*Paragloborotalia mayeri*) have their stratigraphic first or last occurrence within the studied interval. All recorded
taxa were grouped according to their environmental preferences following established environmental assignments
of either mixed layer taxa, open ocean thermocline taxa, open ocean sub-thermocline taxa, upwelling taxa, or
unknown (Table 2).
Through the studied interval, thermocline species and mixed layer taxa are the most abundant (abundance reaches
more than 50%). Both mixed layer and upwelling taxa increase in prevalence through the studied interval, while
thermocline species decrease. A sharp drop in thermocline taxa occurs between 11 Ma and 10 Ma, corresponding
to the disappearance of *Paragloborotalia mayeri*, the dominant taxa until that time. Mixed layer taxa remain at a
near-stable level from 11 Ma onwards. Upwelling taxa are not represented in two samples between 11 Ma and
10.8 Ma, after which this group exhibits a steady increase until the end of the studied interval. Sub-thermocline
taxa are present between 9.0 Ma and 9.5 Ma and account for only a small fraction (less than 3% at most)
of the assemblage.

## 5. Discussion

### 5.1. Definition of taphogroups and their paleoenvironmental significance

Based on the above results, we interpret the analyzed samples in the context of their taphogroups. Taphogroups represent the total preserved fossil assemblage deposited at a given time in the past. Samples assigned to contain the same taphogroup can therefore be assumed to reflect similar local surface water conditions at Site 722.

*Taphogroup 1a*: TG1a is dominated by small reticulofenestrids. We, therefore, interpreted this TG as indicative of high nutrient levels facilitating the proliferation of small bloom-forming placoliths (primarily *Reticulofenestra minuta*; see Table 1). Small reticulofenestrids are commonly associated with high terrigenous nutrients in near-shore environments (see references in Table 1). However, as Site 722 was always located in the open ocean  and sedimentological data (Bialik et al., 2020a) does preclude a change in terrigenous nutrient sources, a different mechanism must be invoked for this dominance of small reticulofenestrids. Studies based on coccolithophore cultures indicate that the proliferation of small placoliths may result from nitrogen limitation in a highly productive open marine environment. For example, Paasche (1998) showed that modern-day coccolithophores tend to increase the formation of small placoliths during N-limitation. Hence, we assume that the proliferation of small reticulofenestrids in the open ocean results from increasing nitrogen limitation compared to other macro- or micronutrients. Such N-limited environemnts often persist in settings with high productivity, due to rapid N-loss during denitrification (Paerl, 2018), which would fit with the above interpretation of small Reticulofenestrid proliferation at Site 722, offshore Oman.

*Taphogroup 1b*: The presence of common *C. floridanus* in combination with abundant small and medium-sized reticulofenestrids within this assemblage indicates elevated nutrient levels, compared to a fully oligotrophic assemblage (see Table 1). The very high but not dominant abundance of small reticulofenestrids may also point to N-limited nutrient sources (see TG 1a). This will be analogous to the fringes of the modern-day Arabian Sea upwelling cell, where nitrogen may be the primary limiting nutrient (Anju et al., 2020), hinting at the presence of more costally confined upwelling during TG1b, which did not fully reach Site 722The overall high diversity, compared to other TGs, suggests that alsooligotrophic conditions may have persited at times (likely sesonally), which may also point towards phosphate co-limitationin at times where upwelling was limited. We thus interpret TG 1b as reflective of open marine conditions with only somewhat elevated nutrient levels compared to an open ocean gyre. Primary nutrient supply, however, is still controlled by nutrients derived through the remineralization of locally produced particulate organic matter (Cullen, 1991), likely supplied to the surface water through seasonal mixing during limited summer monsoons.

*Taphogroup 2*: Within TG 2, common *C. floridanus* occurs together with medium and large reticulofenestrids, commonly associated with warmer water temperature, a deep nutricline, and potentially elevated nutrient conditions. Therefore, we interpret this TG to reflect open marine conditions without directly indicating upwelling-derived nutrients. Nutrients were likely mainly derived through POM remineralization, with low external nutrient influx through upwelling or terrigenous nutrients.

*Taphogroup 3*: Previous studies (Auer et al., 2014; Lohmann and Carlson, 1981) generally associated large reticulofenestrids with high nutrient conditions. Imai (2015) states that dominant large reticulofenestrids

and common discoasterids indicate low nutrient conditions and a deep nutricline compared to a high abundance of small reticulofenestrids.

However, this interpretation is questioned by the association of TG 3 with high TOC, high dust flux, and high silica accumulation rates, indicating strong upwelling conditions (Fig. 4b). Although, similar co-occurrences of diatoms and discoasterids were previously recorded in the eastern equatorial pacific and the Mediterranean (Backman et al., 2013).

While difficult to ascertain, the association of TG 3 with high dust flux and thus additional iron fertilization may represent exceptionally high primary productivity (Guieu et al., 2019). Furthermore, modern analogs based on large *Geophyrocapsa* taxa, descendants of the genus *Reticulofenestra* (Samtleben, 1980; Perch-Nielsen, 1985; Nannotax 3, 2023), are more abundant in high nutrient upwelling zones (Bollmann, 1997). Seasonallity, between summer monsoon and weak or absent winter monsoon however, could also serve to partially address this discrepancy in the interpretation of TG 3 with available environmental data. Diatom and coccolithophore accumulation occur in such a setting in different nutrient regimes. Modern-day culture studies of coccolithophores (Paasche, 1998) also show that the calcification of coccolithophores increases during nitrogen excess and phosphate limitation.

Therefore, we interpret TG 3 as indicative of likely the strongest summer monsoon controlled upwelling for our Middle to Late Miocene study interval. Converesely, a still relatively weak winter monsoon resulted in a deep nutricline during the rest of the year.

*Taphogroup 4a*: Taphogroup 4a is not dominated by a specific reticulofenestrid size range while also containing a diverse assemblage in general (see Table 1). We, therefore, interpret this TG to show weaker upwelling conditions compared to TG3 or TG 1a during transient climatic conditions. Furthermore, weaker productivity is implied by a stronger association of TG 4a with higher Mn/Al values (Fig. 4b).

*Taphogroup 4b*: The high dominance of large reticulofenestrids of TG 4b would suggest elevated, upwelling-derived nutrient levels in a temperate upwelling zone (see TG3 above). Furthermore, the size of experimental studies of calcification rates by Paasche (1998) may also be indicative of p-limitation. High nutrient conditions are corroborated by the general association of TG 4b with siliceous fragments, TOC, and dust flux in the nMDS (Fig. 4b).

*Taphogroup 4c*: Taphogroup 4c is defined by both medium and large reticulofenestrids (Table 1, supplementary material). Therefore, we interpret this TG as indicative of weaker but sustained upwelling conditions. In addition, it shows some association with upwelling indicators such as dust flux and no association with the Mn/Al ratio in the sediment (Fig. 4b), indicating that it only is associated with a overall active upwelling zone and and active Mn-ridirect and therefor OMZ conditions at Site 722.

## 5.2.    Temporal Progression of Environmental Changes

Individual taphogroups represent specific ecospaces, but to understand the relation and transitions between these ecospaces, in their temporal context, their variability has to be considered in relationship to other proxies within a multi-proxy approach. Integrating the analyses of nannofossil taphogroups (Table 1), planktonic foraminifera data (Fig. 5), the abundance of diatom fluxes and geochemical data (Bialik et al., 2020a), we delineate temporal intervals in Site 722. These reflect stratigraphic intervals of specific environmental conditions in the WAS.

**Interval 1 (Base of study interval – 13.4 Ma)**: This interval is characterized by variable taphogroups belonging to TG 1a, TG 2, TG 4a, and TG 4b. The variable taphogroups reflect a diverse and variable nannofloral assemblage

in this interval. Overall the nannofloral assemblages are characterized by an high abundance of *Cyclicargolithus floridanus* (Fig. 5). However, *Cylcicargolithus floridanus* abundances decline through the interval to its stratigraphic Top (T) occurrence at Site 722. In addition, we record abundant small reticulofenestrids and peaks of discoasterids (TG 4a, 4b). The average number of taxa in interval 1 is $14.9 \pm 2.1$ (N = 13), with an average Shannon H' diversity of $1.6 \pm 0.4$.The planktonic foraminifera assemblage is dominated by thermocline-dwelling taxa (predominantly *P. mayeri*). Siliceous fragments are absent. We interpret this interval as a relatively low nutrient environment based on the above multi-group assemblage composition. In particular, the presence of TG 1a and 2 points to only moderately elevated nutrient concentrations in the surface waters at Site 722 during MMCT. The common occurrence of *Sphenolithus* spp. and discoasterids suggests intermitted – potentially seasonal – stratification. These results are consistent with the relatively warm SSTs recorded during this interval (Zhuang et al., 2017), further supporting a generally muted upwelling regime in the WAS during interval 1. These assumptions are corroborated by a more limited OMZ extent in the Indian Ocean, compared to the later Miocene. At Site 722 this is shown declining Mn content. On the Maldives, high Mn concentrations as well as the absence of notable drift deposits, and thus lower wind intensity, also corroborates a generally weaker OMZ during this time (Bialik et al., 2020b; Betzler et al., 2016).

**Interval 2a (13.4 – 12.0 Ma)**: Interval 2a is solely comprised by TG 4c. This taphogroup is characterized by a diverse assemblage with abundant *R. pseudoumbilicus* and common medium-sized reticulofenestrids and discoasterids. The average number of taxa is $16.6 \pm 2.2$ (N = 7), with an average Shannon H' index of $1.8 \pm 0.3$. Siliceous fragments are absent.

Planktonic foraminiferal assemblages are dominated by thermocline species with increased abundances of mixed layer species compared to interval 1. Within interval 2a, a first slight increase in upwelling indicative taxa (primarily *G. bulloides*) is observed. We interpret this interval as indicative of a first shallowing of the thermocline due to the initial strengthening of the wind-driven upwelling regime at Site 722. This intensification is likely related to an intensification of the monsoon system following the end of the MMCT (Betzler et al., 2018). The intensification of the monsoon system is also consistent with the establishment of an increased OMZ extent and drift deposits in the Maldives (Betzler et al., 2016).

**Interval 2b (12.0 Ma – 11.0 Ma)**: Interval 2b comprised primarily of assemblages belonging to TG 4c, with one sample belonging to cluster 1b. The interval similar to interval 2a is characterized by assemblages (TG4c) with abundant medium-sized reticulofenestrids that occur together with an increase in large reticulofenestrids. Furthermore, we detect a low but noteworthy increase in *Umbilicospahera jafari* and a decline in Discoasteraceae. Furthermore, the abundance of small reticulofenestrids is lower than in interval 2a. These differences within the assemblage are also the reason why interval 2 was separated into the two sub-intervals. The average number of taxa in interval 2b is $15.6 \pm 2.6$ (N = 16), with an average Shannon H' index of $1.5 \pm 0.3$. The base of interval 2b also contains the first occurrence of diatoms within the section. Planktonic foraminifera mixed layer taxa decrease noticeably while upwelling taxa further increase in this interval.

We interpret this interval to mark a progressive intensification in the upwelling of high-nutrient subsurface waters. We base this on 1) the increase in siliceous fragments (diatoms and other siliceous biota, 2) higher abundances of upwelling indicative planktonic foraminiferal taxa, 3) generally nutrient-adapted nannofossil taxa (i.e., medium and large sized reticulofenestrids; Beltran et al., 2014; Auer et al., 2015; Imai et al., 2015) show progressive abundance increases. Intensified upwelling is consistent with increasing $\delta^{15}N$ values and continuous cooling at Site 722 (Zhuang et al., 2017; Bialik et al., 2020a). Increased upwelling-derived nutrient access in the northern

Indian Ocean is further supported by increased productivity and nitrogen utilization in the Maldives (Betzler et al.,
2016; Ling et al., 2021). The upwelling intensification after 12 Ma is consistent with an overall increase in global
atmospheric circulation and oceanic current strength, including the Indian Ocean south equatorial current (Fig. 6;
House et al., 1991; Gourlan et al., 2008; Groeneveld et al., 2017; Betzler and Eberli, 2019).
**Interval 3a (11.0 Ma – 9.6 Ma)**: Interval 3a is characterized by a dominance of large reticulofenestrids (*R.*
*pseudoumbilicus*) (TG 3) with intermittently common discoasterids and small reticulofenestrids (TG 4b). Notably,
medium-sized reticulofenestrids show very low abundances compared to the previous intervals. The abundance of
*Umbilicosphaera jafari* is highly variable but overall common, while sphenoliths are rare in the lower part of the
interval before increasing (up to ~ 40 % of the assemblage) in the upper part. Within this interval, we also note the
occurrence of variable abundances of small reticulofenestrids between ~10.5 to 9.9 Ma. The average number of
taxa is $14.3 \pm 5.1$ (N = 22), with an average Shannon H' index of $1.1 \pm 0.4$. The high environmental variability
within this interval is illustrated by alternations between assemblages belonging to TG 3, 4b, and 4c. Diatom fluxes
increase significantly (Fig. 5). Diatoms generally dominate the phytoplankton assemblage, even outcompeting
calcareous nannoplankton in terms of total abundance. High diatom abundances are especially prevalent within
samples assigned to TG 3. Mixed layer taxa dominate planktonic foraminifera assemblages and increase in this
interval, together with upwelling taxa. Notably, thermocline species decline to less than half of their previous
abundance. One sample (722B-34X-3W 30-32) is barren of planktonic foraminifers. The lack of foraminifera is
likely due to the limited sample amounts washed for this study, in conjunction with the high accumulation rates of
phytoplankton (diatoms and calcareous nannofossils) in this stratigraphic interval.
Based on the high abundance of diatoms and a generally high nutrient-adapted nannofossil assemblage, we
interpret interval 3a as a peak in upwelling intensity at Site 722. This interpretation is consistent with previously
published $\delta^{15}$N data from Site 722 and Sites U1466 and U1468, and other geochemical datasets in the Maldives
(Bialik et al., 2020a; Ling et al., 2021). In addition, high productivity and OMZ expansion are further recorded by
heightened TOC, Uranium accumulation, and low Mn deposition within the northwestern Indian Ocean (Dickens
and Owen, 1994, 1999; Betzler et al., 2016; Bialik et al., 2020a). This corresponds to an increase in Antarctic
Bottom Water (AABW) formation due to the expansion of North Atlantic Deep Water (NADW), indicative of an
intensified global thermohaline circulation (Woodruff and Savin, 1989). Increasing numbers of discoasterids in
the upper part of interval 3a, and decreasing diatoms numbers also point towards declining upwelling and, thus,
seasonal nutrient depletion when no summer monsoon-derived upwelling occurs. This pattern of clear seasonality
imparted on the plankton flux further amplifies within the next interval.
**Interval 3b (9.6 Ma – 8.8 Ma)**: Interval 3b continues to exhibit a dominance of large reticulofenestrids (*R.*
*pseudoumbilicus*) (TG 3), although discoasterids noticeably decline and are replaced by higher abundances of
sphenoliths (primarily *Sphenolithus moriformis*), with abundances of ~ 40 % of the total assemblage. Small- and
medium-sized reticulofenestrids are rare in this interval. The average number of taxa is $15 \pm 2.3$ (N = 10), with an
average Shannon H' index of $1.4 \pm 0.3$.
We thus interpret interval 3b to indicate decreasing upwelling intensity based on the increase in nannofossil taxa
adapted to warmer and more stratified water masses, such as *Discoaster* spp. and *Sphenolithus* spp. (Lohmann and
Carlson, 1981; Castradori, 1998; Negri and Villa, 2000; Blanc-Valleron et al., 2002; Gibbs et al., 2004a; Aubry,
2007; Villa et al., 2008; Schueth and Bralower, 2015). The waning upwelling of the northern Indian Ocean is
corroborated by the proliferation of warm water diatom taxa in the Maldives (Site 714; Boersma and Mikkelsen,
1990). Decreasing $\delta^{15}$N values support waning upwelling-derived productivity after 10 Ma at both Site 722 and in

the Maldives and decreasing TOC fluxes at Site 722 (Gupta et al., 2015; Bialik et al., 2020a; Ling et al., 2021). It is, however, important to note that these changes are not reflected in the planktonic foraminifera community, which shows a continuously high presence of upwelling taxa (e.g., *G. bulloides*). One possibility would be that the upwelling cell became more seasonal, with nannoplankton-dominated photoautotrophic communities proliferating seasons with lower upwelling. However, primarily heterotrophic, non-symbiont-bearing taxa such as *G. bulloides* were still sustained by high primary productivity during monsoon season, as is the case in the present-day upwelling cell along the Oman Margin (Schiebel et al., 2004; Rixen et al., 2019b).

We assume that this waning in upwelling is related to a decrease in the hemispheric temperature gradients leading to a weaker summer monsoon wind system in the Indian Ocean. This reduction in temperature gradients is consistent with a decreasing trend in minimum deep-water temperatures, based on global benthic foraminifera compilations and deep-water records from the ninety-east-ridge (Site U1443; Fig. 1) (Lübbers et al., 2019; Westerhold et al., 2020). Furthermore, pollen data (Pound et al., 2012) suggests that progressive cooling of the northern hemisphere (NH) over the Middle to Late Miocene intensified. Northern hemisphere cooling consequently reduced the asymmetry of hemispheric temperature gradients, thereby reducing summer monsoon wind intensity by muted northward migration of the intertropical convergence zone (ITCZ) in NH summer (Gadgil, 2018; Yao et al., 2023).

**Interval 4 (8.8 Ma – top of study interval)**: Interval 4 – consisting of only three samples – is defined by the bloom of small reticulofenestrids (*R. minuta*) in the nannofossil assemblage. We also note an elevated abundance of *Umbilicosphaera jafari* and a marked decline in *Sphenolithus* spp. relative to interval 3b. This interval consists entirely of assemblages belonging to TG 1b. The average number of taxa is $17.3 \pm 0.5$ (N = 3), with an average Shannon H' index of $0.5 \pm 0.0$. Despite the high number of nannofossil taxa in this interval, the low diversity directly results from the dominance of small reticulofenestrids. Siliceous fragments (primarily diatoms) persist but are much rarer than in interval 3. This reduction in diatom fluxes is part of an ongoing decrease in biogenic silica accumulation at Site 722, which culminates in a shift from phytoplankton to zooplankton-dominated silica accumulation by ~8 Ma (Nigrini, 1991; Prell et al., 1992). Planktonic foraminifera assemblages remain consistent with the upper part of interval 3, showing relatively high abundances of upwelling and mixed-layer taxa. We interpret this interval as a new nutrient regime which likely led to a significant turnover in coccolithophore species around the same time (Young, 1990; Imai et al., 2015). However, the low sample number in this interval limits further interpretation.

### 5.3.    Plankton community responses paleoenvironmental changes

Based on the intervals defined by the nannofossil taphogroups, a progression of plankton communities becomes apparent within the Middle to Late Miocene at Site 722. Their variation highlights the strong interactions between monsoon wind strength, nutrient availability, and primary productivity. Therefore, we link our new assemblage data with an extensive data compilation highlighting a progressive upwelling increase, which leads to thermolcine shoaling. This thermocline shoaling, in turn, results in declining sea surface temperatures and increased surface water productivity through the upwelling nutrient-rich thermocline waters along the Oman Margin during this time (Fig. 3; Zhuang et al., 2017; Bialik et al., 2020a).

Declining high Mn/Al ratios and diverse nannofossil assemblages point towards a relatively low nutrient regime between 15.0 and 13.5 Ma. Patterns of Mn decline have been observed since at least 15 Ma in the Maldives, which is in line with observations at Site 722 (Betzler et al., 2016; Bialik et al., 2020a, b). This period thus represents a

progressive increase in upwelling intensity during the MMCT due to globally declining SSTs and sea levels following the end of the MCO (Zhuang et al., 2017; Miller et al., 2020). Both nannoplankton and planktonic foraminifera reflect primarily open marine, low-nutrient conditions (Sexton and Norris, 2011; Lessa et al., 2020). By 13.5 Ma, these progressive changes culminate in a first sustained community shift in both nannofossil and planktonic foraminifera records (Figs. 2 & 5).

We consider these shifts to be a coupled response of Site 722 phytoplankton communities to increased surface water nutrient levels that subsequently allowed a population increase of heterotrophs such as foraminifera. These changes are consistent with establishing a more pronounced upwelling regime, which also resulted in the expansion of the OMZ further into the Indian Ocean, reaching the Maldives by ~13 Ma. Furthermore, available TOC data still show low accumulation rates at Site 722 at this time, indicating that organic matter was still recycled mainly within the expanding OMZ (Bialik et al., 2020a).

By ~12 Ma, another phytoplankton community shift (see interval 2b) leads to a size increase in the reticulofenestrids, lower nannoplankton diversity, and a higher abundance of thermocline-dwelling planktonic foraminifer taxa (Fig. 5). Together with increasing TOC fluxes (Fig. 3), all these shifts point towards increasing productivity. These changes, however, happen without any significant changes in overall temperature within the upwelling zone (Zhuang et al., 2017). A northward shift of the southern hemisphere westerlies is recorded by 12 Ma (Groeneveld et al., 2017). We hypothesize that this shift and a potential increase in wind strength may have also increased nutrient concentrations in intermediate water masses within the sub-Antarctic frontal system. This interpretation would be in line with the effect increasing sea ice cover would have had on intermediate water nutrient concentrations based on modelling data and evidence from southern hemisphere records (Sarmiento et al., 2004; Sarmiento and Gruber, 2013; Laufkötter and Gruber, 2018; Groeneveld et al., 2017). Such enhanced nutrient transport within the thermocline would reconcile increased productivity without increasing the total volume of upwelling – and consequently reducing SSTs - along the Oman Margin. The first occurrence of diatoms within this interval may also point towards a shift in nutrient availability and increased phosphorus and silicon availability within the upwelling cell and likely globally (Keller and Barron, 1983). Decreasing P- and Si-limitation would thus provide more favourable conditions for highly efficient photosynthesizers, such as diatoms within the water column (Schiebel et al., 2004; Brembu et al., 2017; Sarmiento and Gruber, 2013). Within the plankton community, we also note the first intermittent occurrences of elevated *G. bulloides* abundances, indicative of high productivity upwelling conditions (Kroon et al., 1991; Gupta et al., 2015).

By 11 Ma, global climatic shifts and further decreasing sea levels (Miller et al., 2020; Westerhold et al., 2020) led to another step in the water masses upwelling in the WAS (Fig. 6). As a result of these water mass changes, diatoms dominate our phytoplankton record by 11 Ma, outpacing nannoplankton for the first time, while we note a first sustained occurrence (> 25 %) of *G. bulloides*. Therefore, we interpret this shift as the inception of sustained primary productivity within the upper water column of an upwelling cell supplied with enough Si, as well as P and N, to sustain a large diatom population (Brzezinski, 1985; Sarmiento and Gruber, 2013; Closset et al., 2021).

However, the abundance of discoasterids and sphenoliths within our nannofossil record (Fig. 5) still needs to be reconciled with this interpretation. Both taxa are considered to be indicative of low nutrient conditions and increased stratification (Gibbs et al., 2004a; Schueth and Bralower, 2015; Karatsolis and Henderiks, 2023). This information is thus contrary to our recorded high abundances of mixed layer dwelling foraminifera and high nutrient-adapted diatoms dominating the phytoplankton record. A possible way of integrating these opposite

requirements is to evoke a highly seasonal upwelling cell with strong upwelling in one season and calm and
stratified surface waters providing a deep thermo- and nutricline in the other.

This seasonal variability is most evident after 9.6 Ma when *Sphenolithus* abundances also increase together with
overall nannofossil diversity (Fig. 5, interval 3b). These changes in the nannofossil community are also associated
with decreasing diatom abundances and TOC fluxes while upwelling indicative planktonic foraminifera taxa
remain common. It thus seems that an initial spike in upwelling and, therefore, diatom accumulation waned again,
pointing towards a significant reorganization of the upwelling cell after 9.6 Ma.

Within the topmost three samples of the record, belonging to interval 4, we note an increase in small
reticulofenestrids corresponding to the base absence of *Reticulofenestra pseudoumbilicus* around 8.8 Ma,
according to accepted nannofossil biostratigraphy (Young, 1990; Backman et al., 2012; Imai et al., 2015). We note
that this significant size change and an increase in small placoliths are very pronounced within our WAS records
from Site 722, in agreement with Young (1990). While we cannot contribute to the discussion of whether this
assemblage shift constitutes an evolutionary-driven adaptation of taxa within the genus *Reticulofenestra* or purely
an ecophenotypically driven size adaption (Young, 1990; Imai et al., 2015). We still note that a clear link to
changing nutrient levels within the upwelling cell is becoming apparent. Imai et al. (2015) further hypothesized
that the size shift is related to nutrient increases within the Indo-Pacific. Based on our records of high nutrient
conditions and likely at least intermittent seasonal eutrophication persisting from at least 11 Ma, we cannot
completely follow their hypotheses that increasing nutrient levels within the surface ocean were the sole driver for
this size shift. Therefore, we propose that changing nutrient limitation within the mixed layer may have played an
important, as-of-yet unconsidered role in defining the predominant assemblage structure within the WAS
upwelling system during the Middle and Late Miocene (Fig. 7).

## 5.4.    Contextualizing the primary drivers for plankton community shifts

The modern productivity patterns and oxygen depletion in the northwestern Indian Ocean differ significantly from
those observed in the studied period. For example, the increase in Mn content in the Maldives in the Pliocene
(Betzler et al., 2016) suggests a significant reduction in Mn redirection, which continued until today. This is indeed
visible in present-day oceanographic records, where elevated Mn concentrations are only found near the margins
of the Arabian Sea (ThiDieuVu and Sohrin, 2013). Meanwhile, denitrification in the Eastern Arabian Sea appears
to have only become significant during the Pliocene (Tripathi et al., 2017). These changes in productivity patterns
thus may indicate that the WAS was potentially more productive during the Late Miocene than today and
potentially even supported an expanded OMZ (Dickens and Owen, 1999, 1994).

Despite that, we note that even in the most productive parts of the Arabian Sea, conditions are rarely eutrophic
(Fig. 1a). As such, ascribing permanent eutrophic or even mesotrophic conditions to any of these assemblages is
unlikely to be reasonable. On the other hand, nannofossil assemblages such as TG 3 with combined diatom
occurrences possibly indicate the prevalence of mesotrophic and eutrophic conditions. Diatoms are generally less
adapted to low nutrient levels, requiring much higher P and N levels to form blooms compared to coccolithophores
(Hutchins and Bruland, 1998; Litchman et al., 2006). If enough nutrients (including Si) are available, they tend to
outcompete coccolithophores quickly and begin to dominate the mineralizing phytoplankton community (Schiebel
et al., 2004; Brzezinski, 1985; Closset et al., 2021). Based on modern analogs, it seems likely that shifts in the
nutrient content of upwelling waters may have played and important role in controlling the observed patterns in
the plankton community along the WAS during the Middle to Late Miocene. In particular after 13 Ma, where a

stustained and stable SAM regieme seems to have existed during the northern hemisphere summer (Betzler et al.,
2016). To disentangle these patterns we therefore focus on understanding observed patterns of the two dominant
phytoplankton groups present within our record, with the context of their ecological preferences and primary
nutrient requirements within our study interval.
The co-occurrence of diatoms, discoasterids, and sphenoliths in the upper part of the studied interval (Fig. 5) thus
suggests that while nutrient levels were high, upwelling was likely highly seasonal. For the WAS, high seasonality
may be the result of strengthening summer monsoon winds with no changes in winter monsoon winds (Schiebel
et al., 2004; Rixen et al., 2019b; Sarr et al., 2022). Increasing summer but stable or absent winter monsoon
conditions are likely the result of increased cooling in the southern hemisphere (Bialik et al., 2020a; Gadgil, 2018;
Sarr et al., 2022). This asymmetric cooling strengthened the summer monsoon compared to the winter monsoon
system, which only intensified ~7 Ma (Gupta and Thomas, 2003; Holbourn et al., 2018; Rixen et al., 2019b).
The variability in wind and upwelling intensity and their interaction with nutrient availability, thus, likely also
affected the community structure and size variability of primary producers on longer geological time scales. The
community structure of primary producers then exerted control on first-level consumers, such as planktonic
foraminifera.
Upwelling-derived TOC accumulation, primary productivity assemblages, and upwelling indicative foraminifera
show distinctive patterns, which are, however, not in complete agreement with wind proxies and the suggested
expansion of the OMZ around 13 Ma (Betzler et al., 2016). These discrepancies resulted in a long-standing debate
about the validity and usefulness of upwelling proxies as monsoonal indicators (Betzler et al., 2016; Clift and
Webb, 2018; Bialik et al., 2020a; Yang et al., 2020; Sarr et al., 2022). We propose that this disagreement is
primarily due to inadequate treatment of nutrient limitation and nutrient supply in conjunction with wind speed
when evaluating primary productivity in the WAS (Fig. 5, 7).
Modern-day upwelling zones in the low-to-mid-latitudes are generally well supplied in macro-nutrients, resulting
in iron-limited environments or other micro- and nano-nutrient limitations (Moore et al., 2013). However,
currently, the fringing areas of upwelling zones are commonly N-limited through increased denitrification in
underlying OMZs (Moore et al., 2013; Bristow et al., 2017; Anju et al., 2020; Buchanan et al., 2021; Ustick et al.,
2021; Buttay et al., 2022). Within the WAS upwelling zone, major nutrients N, P, and to some degree, minor
nutrients such as Si are replenished through local recycling and intermixing with deep and intermediate water
masses originating from Antarctica (Fig. 7; Sarmiento et al., 2004; Meisel et al., 2011; Sarmiento and Gruber,
2013; Laufkötter and Gruber, 2018). Iron, a key micronutrient, is primarily supplied through dust and riverine
influxes from surrounding continental sources (Kunkelova et al., 2022; Moore et al., 2013; Guieu et al., 2019).
Accepting that the wind regime had reached peak intensity by 13 Ma following a gradual increase from the end of
the MCO (Betzler et al., 2016, 2018), the significant increase in diatom abundance and TOC accumulation after
12 Ma is not contemporary. Therefore, the availability of nutrients and the nutrient composition also played a key
role in defining the variability between coccolithophore and diatom abundances within the WAS upwelling cell.
Moreover, the shift in the reticulofenestrid morphotypes (Fig. 5) may also be linked to the state of nutrient
limitation. Paasche (1998) also has shown that modern-day coccolithophores tend to increase the formation of
small placoliths during N-limitation.
Therefore, the shift towards higher primary productivity after 12 Ma, including first record of diatoms at Site 722,
may indicate a change in nutrient composition along the WAS without necessitating a change in monsson wind
strength. Notably, during this time, the northward expansion of the southern hemisphere westerlies shifted the

position of the polar and sub-Antarctic frontal system (Fig 6). In particular, the Late Miocene sea ice expansion after 11 Ma strongly affected the Antarctic frontal system and, in turn, the nutrient enrichment of intermediate waters formed in this region (Groeneveld et al., 2017; Bijl et al., 2018; Laufkötter and Gruber, 2018). Here we propose that changes in the mode of intermediate water formation significantly increased the nutrient availability in intermediate waters in the Antarctic frontal system, resulting in modern-like downwelling dynamics around Antarctica (Fig. 7). Furthermore, many modeling studies support the assumption that climatic changes affecting the Antarctic frontal system can strongly influence global productivity patterns (Sarmiento et al., 2004; Laufkötter and Gruber, 2018; Moore et al., 2018; Taucher et al., 2022). We, therefore, propose that the Middle to Late Miocene productivity changes in the WAS offer compelling evidence for this hypothesis.

## 5.5.  Synthesizing Miocene nutrient transport and monsoonal upwelling

Thus far, the discussion was focused on local aspects of the record in Site 722 in the WAS and northwestern Indian Ocean. However, the interconnected nature of the oceanic circulation and nutrient rejuvenation system means that critical mechanisms may be overlooked without a global perspective. For example, modeling evidence for nutrient transport and nutrient enrichment in low-latitude upwelling cells allows for the construction of a timeline of changes along the WAS and their interaction with plankton communities. Moreover, a complete oceanic perspective allows for contextualization into the broader evolution of the ocean-atmosphere system.

Initial plankton community structures agree with a generally low nutrient regime in a somewhat muted wind regieme, based on a large amount of deep thermocline dwelling taxa in the foraminifera community, likely following the dominant phytoplankton primary productivity in the deeper photic zone (Lessa et al., 2020). In addition, the mixed layer is dominated by a diverse nannofossil assemblage (H'-diversity of around 1.5 within intervals 1 and 2). During the MMCT, wind shear strengthened by 13 Ma, resulting in a significant global shift in ocean-atmospheric circulation exemplified in the global reorganization of carbonate-platform geometries and thermocline deepening and ventilation at Site 722, as shown by the increase in mixed-layer dwelling planktonic foraminifera (Betzler et al., 2016, 2018; Betzler and Eberli, 2019; Lessa et al., 2020).

Modeling studies for the WAS link the initial intensification of upwelling and wind shear to a combination of increased latitudinal temperature gradients and the emergence of the Arabian Peninsula during the Middle Miocene (Zhang et al., 2014; Sarr et al., 2022; Yang et al., 2020). Notably, while OMZ expansion and Mn redirection are evident since at least ~14 Ma at Site 722 (Bialik et al., 2020a), available productivity records support at most intermittently mesotrophic and likely P- and N-limited conditions before ~12 Ma (Fig. 5). We thus propose that the upwelling cell in the WAS was wholly influenced by strong post-MMCT winds by 13 Ma. Productivity was still limited by the upwelling of comparably low nutrient intermediate waters of local origin (Fig. 7). Likely originating in the marginal seas of the northwestern Indian Ocean, these water masses may have been remnants of the Tethyan Intermediate Water (TIW). While the Tethyan Seaway had terminated between 14 and 15 Ma (Bialik et al., 2019), TIW or a similar high salinity mass (Woodruff and Savin, 1989; Smart et al., 2007) was still affecting the Northern Indian Ocean until at least 12 Ma. This remnant TIW can be considered a more potent form of the modern Red Sea and Persian Gulf Intermediate Waters (RSPGW; Fig 7). These warm and salty intermediate waters may have played a much more substantial role in the WAS during the early stages of the uplift of the Arabian Peninsula (Woodruff and Savin, 1989; Tomczak and Godfrey, 2003; Chowdary et al., 2005; Smart et al., 2007; Acharya and Panigrahi, 2016). The influence of remnant TIW would also align with the high abundance of

thermocline-dwelling taxa until 12 Ma, which we infer to be representative of a shallow and/or a poorly ventilated thermocline (Sexton and Norris, 2011; Lessa et al., 2020).

It thus seems likely that late Middle Miocene WAS upwelling may have been relatively nutrient-poor. We speculate that these water masses may have suppressed primary productivity, muting the influence of the increasing Findlater Jets and the emerging Arabian Peninsula (e.g., Sarr et al., 2022) compared to today. Invoking significant TIW upwelling until at least 12 Ma would further reconcile the discrepancy between the occurrence of drift deposits in the Maldives, and thus strong monsoon winds and the first clear evidence for strong upwelling in the WAS, with the abundance increase of upwelling indicative planktonic foraminifera (e.g., *G. bulloides*; Fig 5) and the first occurrence of diatoms at Site 722 (Fig 5; Kroon et al., 1991; Huang et al., 2007b; Gupta et al., 2015; Bialik et al., 2020a). This change in nutrient availability is also reflected by a contemporary increase in medium-sized reticulofenestrids (*R. antarctica* and *R. haqii*), which are generally assumed to reflect higher nutrient availability due to upwelling (Fig. 5; Auer et al., 2019 and references therein).

Productivity in the WAS thereby only began to increase as remnant TIW got progressively supplanted by other, more nutrient-rich, water masses. At present, the waters upwelling in the Arabian Sea is primarily regarded to be ICW, which therefore also includes IIW, SAMW and AAIW (You, 1997, 1998; Böning and Bard, 2009; Munz et al., 2017; Chinni and Singh, 2022). Today AAIW and SAMW forming in the northern branch of the Antarctic Divergence, control up to 75% of low-latitude productivity (Sarmiento et al., 2004). We hypothesize that the increasing formation of AAIW and SAMW following the northward shift of the westerlies around 12 Ma (Fig.6) may have modulated low latitude productivity (Groeneveld et al., 2017; Laufkötter and Gruber, 2018; Moore et al., 2018; Taucher et al., 2022). This time also aligns well with the proposed inception of the northward shift of southern hemisphere climate belts and the invigoration of the south equatorial current (LeHouedec et al., 2012; Reuter et al., 2019). Following that, it can also be assumed that by 12 Ma, the northward expansion of the southern hemisphere Westerlies resulted in a near-modern Antarctic Divergence (Groeneveld et al., 2017; Laufkötter and Gruber, 2018).

This global change in circulation patterns was fully established by 11 Ma, with cool nutrient-rich SAMW/AAIW waters reaching Site 722, evidenced by a further SST drop (Zhuang et al., 2017). This resulted in the highest productivity in the WAS upwelling cell during the Miocene (Figs. 5-7). The Late Miocene high-productivity interval in the WAS, is thus the result of intense summer monsoon-dominated AAIW/SAMW upwelling, fueled by the Findlater Jets and forced by steep latitudinal temperature gradients and favourable tectonic conditions on the Arabian Peninsula (Pound et al., 2012; Zhang et al., 2014; Sarr et al., 2022). Summer months were thus characterized by eutrophic P-, N-, and potentially Si-enriched waters, allowing the proliferation of diatoms and other siliceous organisms. Winter months, in contrast, favoured the accumulation of deep-dwelling discoasterids that utilized the nutrient-rich waters below a relatively deeper winter thermocline. Higher abundances of mixed-layer dwelling taxa also reflect the increased mixed-layer depth (Fig. 5). Expanding AAIW/SAMW-fueled high productivity that consequently also resulted in the highest recorded TOC fluxes between 11 – 10 Ma and a substantial OMZ expansion deep into the equatorial Indian Ocean (Dickens and Owen, 1994; Bialik et al., 2020a). Increasing OMZs also resulted in a global increase in denitrification, which is well-recorded in foraminifer-bound $\delta^{15}N$ records, showing a trend from more oxygenated intermediate waters during the MCO to lower oxygenated waters in the Late Miocene in the Indo-Pacific (Auderset et al., 2022).

By 10 Ma, OMZs had reached a critical threshold, leading to another substantial change in nutrient conditions within the WAS upwelling. Through increased denitrification in the OMZ underlying the upwelling cell, nitrate

and ammonia were lost through bacterial conversion to $N_2$ (Sigman and Fripiat, 2019). Strong denitrification subsequently led to increasingly N-limited water masses upwelling within the WAS. Although concrete evidence is only presented for the WAS, these patterns could also have occurred globally, considering the clear evidence for decreasing ocean oxygenation during the Late Miocene (Auderset et al., 2022). The Late Miocene N-limitation in the WAS upwelling cell is chiefly expressed by a decline in diatom abundances after 10 Ma, in conjunction with overall community shifts in the nannofossil assemblage.

Total upwelling intensity also remained consistently high, as indicated by the available SST record of Zhuang et al. (2017). Primary productivity thus remained relatively high, which is characterized by the continued presence and even dominance of large reticulofenestrids, diatoms, and the continuously high TOC concentration within the sediment (often > 1 wt.%; Fig. 3). We thus assume that the drop in diatom abundance and intermittent decline in $\delta^{15}N$ values at Site 722 (Figs. 3, 5.) were not caused by decreasing upwelling intensity but rather a decline in P and Si availability and, thus declining export of diatom-derived organic matter. The increase in sphenoliths within our Site 722 record (Fig. 5) could indicate increased environmental stress within the nannofossil assemblage (Wade and Bown, 2006). Sphenoliths are likely not a good indicator of long-term stratification changes (Karatsolis and Henderiks, 2023) in highly seasonal upwelling regimes like the WAS, as high TOC and thus sustained, but lower, diatom fluxes indicate continued upwelling after 10 Ma at Site 722. Sustained seasonal upwelling and high organic matter export (Fig. 3) are further inferred by decreasing organic carbon $\delta^{13}C$ throughout this interval (Bialik et al., 2020a and references therein).

By 8.8 Ma, the adaption of smaller reticulofenestrids may result in an evolutionary adaption to the continued N-limited nutrient availability in the WAS. We base this interpretation on the nutrient adaption of coccolithophorids based on modern culture experiments (Paasche, 1998). Although somewhat anecdotal, these offer the currently best explanation to reconcile the recorded history of Site 722 upwelling changes with the stark shifts in reticulofenestrids size ranges. It should be noted that these shifts have been recorded throughout the mid- and low latitudes of the Indopacific (Young, 1990; Imai et al., 2015). However, the full impact of this hypothesis needs to be tested further.

The data compilation of Young (1990) further shows that the recorded Late Miocene size shift was primarily limited to the low and mid-latitudes, with larger reticulofenestrids persisting within in the higher latitudes. We propose that the transition in *Reticulofenestra* morphology from large to small morphotypes thus primarily represents a significant shift in nutrient limitation rather than total nutrient availability within the mid to low latitudes. We further argue that this turnover reflects N-limitation within the low- and mid-latitudes due to sustained and intense denitrification after 12 Ma (Auderset et al., 2022). Further studies, particularly on ultrastructural morphotaxonomy of reticulofenestrids, will be needed to fully disentangle the implications of the proposed N-limited nanno-floral turnover.

The highly opportunistic small *Reticulofenestra* morphotype was subsequently also able to sustain phytoplankton blooms in the WAS, as evidenced by the significant increase in nannofossils within the sediment (Fig. 5). Furthermore, the high mass of small coccolith cells potentially also contributed to the re-establishment of strong denitrification as evidenced by a rise in $\delta^{15}N$-values after 8.8 Ma (Fig. 3), as their additional biomass contributed to OMZ re-expansion. Detailed records of Late Miocene OMZ strength throughout the Indian Ocean will, however, be necessary to fully quantify the impact on local upwelling. Local tectonics also began to modify the region configuration at this time (Rodriguez et al., 2014), leading to bottom current intensification (Rodriguez et al., 2016), which may have also modulated subsequent OMZ dynamics (Dickens and Owen, 1999).

## 6. Conclusions

We present fully quantitative nannofossil and planktonic foraminifera assemblage data in conjunction with diatom frustule abundances for Site 722. Within a multi-proxy framework, these novel data allowed us to disentangle the complex and long-debated changes within the upwelling system of the WAS in the Middle to Late Miocene. We show that the Findlater Jets, and thus Indian summer monsoon wind strength, are the primary drivers of upwelling. However, wind-driven upwelling is also clearly modulated by local and global water mass changes and changing nutrient fluxes. In particular, changing nutrient transport through intermediate waters has had a significant – until now unconsidered – impact on primary productivity patterns and plankton communities over the Middle and Late Miocene in the Indian Ocean. We, therefore, reach the following key conclusion:

(1) the expansion and evolution of upwelling within the WAS as a complex interplay of regional tectonics, global climate, and ice volume changes affected upwelling intensity and nutrient availability. The present study emphasizes that wind and nutrient changes are intrinsically related but do not necessarily operate in tandem on longer supra-Milankovitch time scales. It is, therefore, crucial to consider both water mass changes and atmospheric conditions when investigating past wind-driven upwelling regimes.

(2) The interaction first invigorated monsoonal circulation after the MMCT before resulting in the reorganization of intermediate water circulation, controlled by the inception of a near-modern configuration of the Antarctic Divergence, which supplied nutrient rich intermediate waters to the low laitutdes.

(3) These processes led to the progressive establishment of near-modern nutrient transport within the Indian Ocean by 12 to 11 Ma. Furthermore, these changes acted together with denitrification in expanding global OMZs (Auderset et al., 2022) to increase N-limitation and subsequent adaption of coccolithophorids to the new nutrient conditions in the mid to low latitudes.

(4) We provide a timeline of events that agrees with global climatic and local productivity patterns, which are all linked through the invigoration of upwelling cells and nutrient fluxes through intermediate water masses into the lower latitudes. In particular past changes in intermediate water mass circulation, replenishment, and expansion appear to be a key – and critically understudied – aspect within paleoceanography and paleoclimatology that is crucial to understanding past and, thereby, future low latitude productivity.

## 7. Data and code availability

Data and code are available from the supplementary material and on Pangaea (DOI: will be provided once available).

## 8. Author contribution

**GA:** designed the study, acquired funding, conducted nannofossil counts and statistics, wrote the first draft, edited the text, and drafted the figures. **OMB**: designed the study, performed statistical analyses, wrote the first draft, edited the text, and helped draft the figures. **MEA**: performed planktonic foraminifera taxonomic analysis and assemblage interpretation and contributed to the first draft of the text. **NVV**: helped draft the figures and contributed to data interpretation, edited the final draft of the MS. **WEP**: supervised and conducted foraminiferal analysis and contributed to writing and editing of the text.

## 9. Competing interests

The authors declare that they have no conflict of interest.

## 10. Acknowledgments

This research used samples and data provided by the Ocean Drilling Program (ODP) and the International Ocean Discovery Program (IODP).  This study was funded by the Austrian Science Fund (FWF Project P36046-N; MIO:TRANS – Nutrient Fluxes in the Miocene Indian Ocean). OMB is partially supported by  the German (GEOMAR)-Israeli (University of Haifa) Helmholtz International Laboratory -The Eastern Mediterranean Sea Centre- An Early-Warning Model-System for our Future Oceans: EMS Future Ocean Research (EMS FORE). Furthermore, the authors would like to thank all Bialik et al. (2020) authors for their invaluable contribution to this research and their expertise in interpreting the data. In particular, we would like to thank Dick Kroon for his early support of these studies and his invaluable discussions on the subject matter.

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

1423

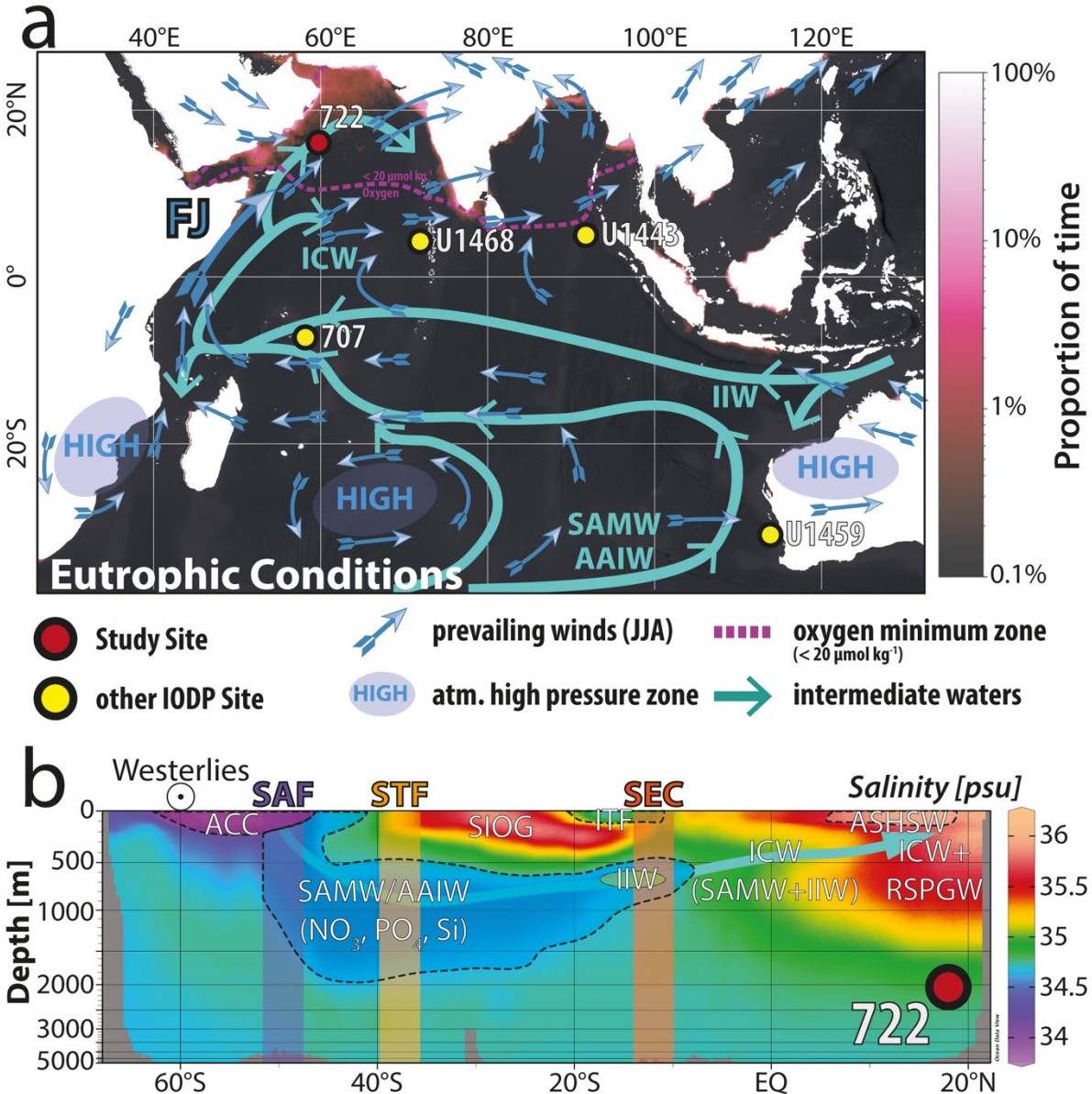

1424

Figure 1: a) Location map showing the study site ODP Site 722 and IODP Site U1468 and the prevalent summertime wind patterns following Bialik et al. (2020a). Generalized flow flow-paths of dominant intermediate waters of the indian Ocean follow You (1998) and Böning (2009), The present-day extent of the oxygen minimum zone is shown as a pink dashed line denoting oxygen concentrations < 20 µmol kg-1 at a water depth of 200 m (McCreary et al., 2013; Garcia et al., 2018). Eutrophication (magenta shading) data was provided by the E.U. Copernicus Marine Service Information using the Global Ocean Colour (Copernicus-GlobColour), Bio-Geo-Chemical, L4 (monthly and interpolated) from Satellite Observations (1997-ongoing); https://doi.org/10.48670/moi-00281. Shading represents gap-filled daily Chlorophyll-a product of Copernicus GLobColour L4 (Gohin, 2011; Hu et al., 2012; Garnesson et al., 2019) and indicates the proportion of time spent in eutrophic conditions in the region, based on the proportion of days (1998-2022) where Chlorophyll-a concentration exceeded a threshold of 7.3 mg m-3 (derived from Carlson, 1977). The python code used to generate the base map is available in the supplementary material; b) Salinity profile generated based on the world Ocean Atlas 2018 salinity data (Zweng et al., 2019) through the Indian Ocean from 65°S to 20°N. The plot was generated using Ocean Data View (Schlitzer, 2021). Water masses are differentiated based on their salinity signature outlined with dashed lines and labeled. Furthermore major frontal systems and currents are also indicated. Abbreviations: Antartic Intermediate Water (AAIW), Antartic Circumpolar Current (ACC), Arabian Sea High Salnity Water (ASHSW), Indian Central Water (ICW), Indonesian Intermediate Water (IIW), Red Sea/Persian Gulf Water (RSPGW), sub-Antarctic Mode Water (SAMW), Southern Indian Ocean Gyre (SIOG),


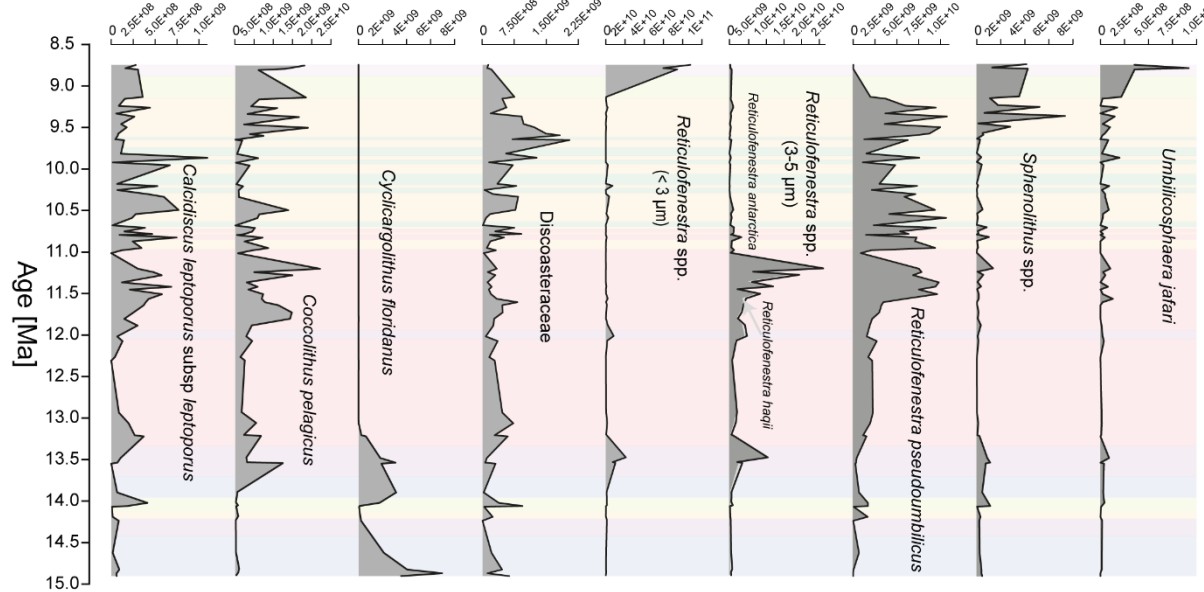

**Figure 2: Abundance data of key nannofossil taxa presented as numbers per gram of carbonate over the study interval**
**following the methods of Bordiga et al. (2015). The used age model is based on Bialik et al. (2020a). Medium-sized**
**reticulofenestrids are separated into morphotypes with an open central area (Reticulofenestra haqii) and a closed**
**central area (R antarctica). Discoasteraceae include the genera Discoaster and Catinaster. Color coding represents the**
**cluster assignment based on the nannofossil assemblage shown in fig. 4a.**

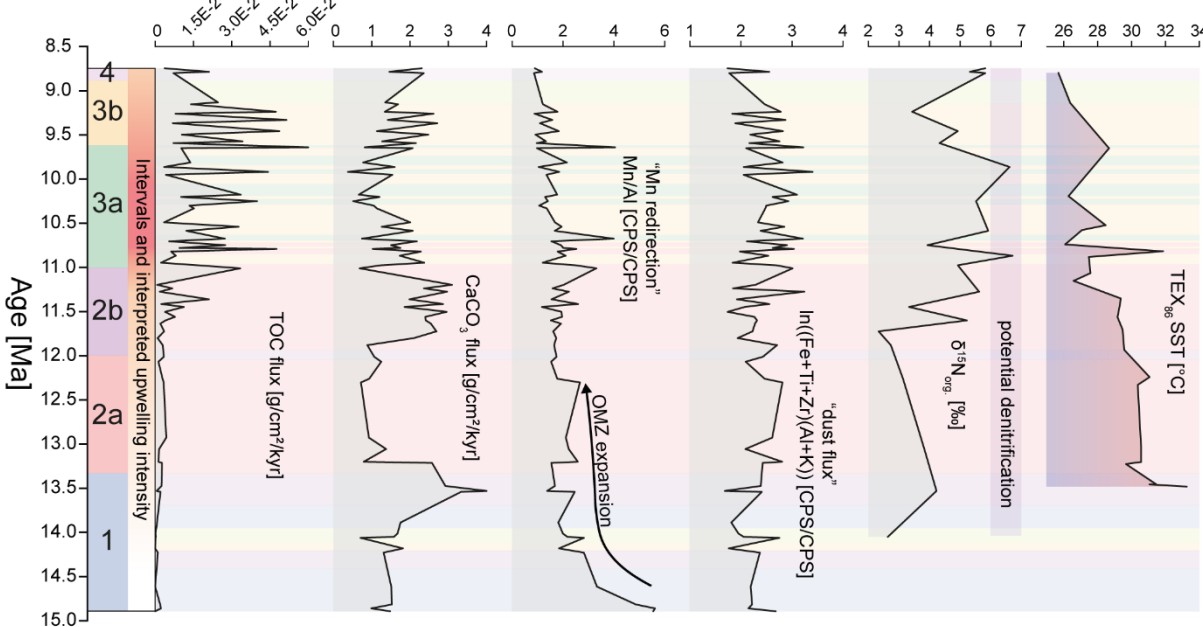

**Figure 3: Geochemical data initially published by Bialik et al. (2020a) as well as TEX$_{86}^{H}$ based SST data of Zhuang et al.**
**(2017). Data is shown in conjunction with the cluster analysis results based on the nannofossil assemblages, as shown in**
**figure 4a. Total organic carbon (TOC in wt.%) is based on bulk sediment measurevments. The Mn/Al ratio and the**
**shown dust flux proxy, are based on benchtop XRF counts. Dust flux is calculated as ln((Zr+Ti+Fe)/(Al+K)) based on**
**Kuhnt et al. (2015), with higher values indicating higher deposition of dust-born minerals at Site 722. Nitrogen isotopic**
**data indicate increasing denitrification of sinking organic matter with higher values. On the left of the figure we also**
**show intervals 1 – 4 and their respective sub-intervals a/b and the resulting interpreted upwelling instensity. All data is**
**unterpinned by the assigned clusters as defined in Figure 4.**


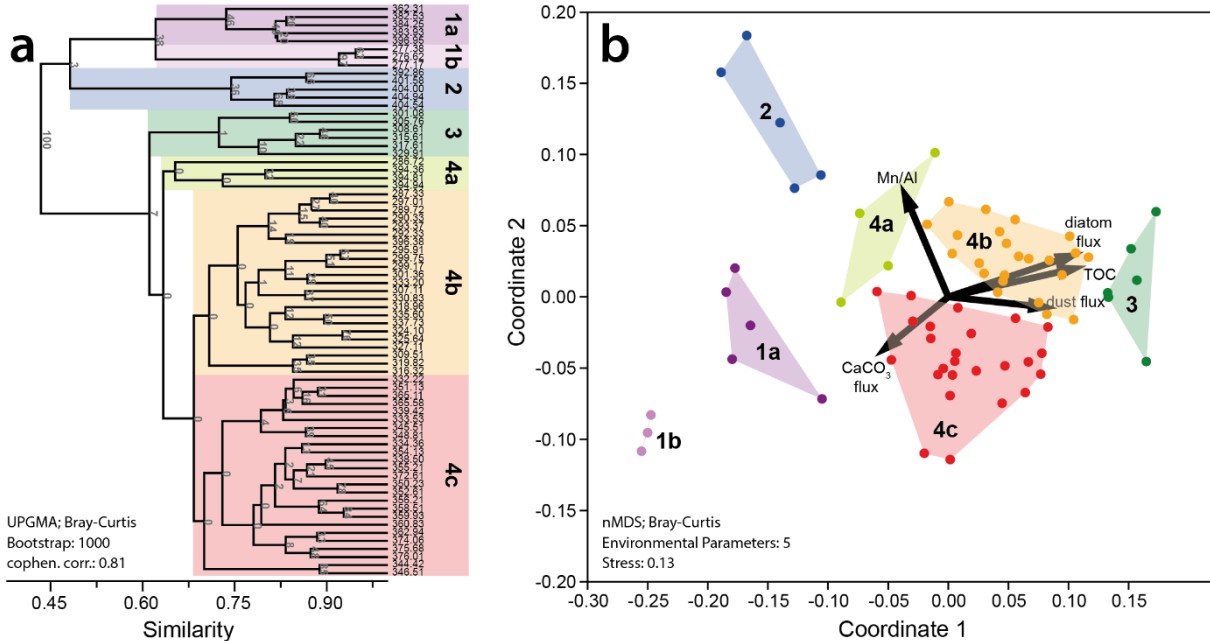

Figure 4: Cluster analysis (a) and nMDS (b) based on the datasets shown in figs. 2 and 3. The geochemical data serves as paleoenvironmental proxies for high productivity (total organic carbon and siliceous fragments), high wind intensity (dust flux), water column oxygenation (Mn/Al), and high carbonate accumulation (CaCO3 flux). Note the high correspondence of clusters 3 and, to some degree, 4b diatom accumulation, dust flux, and high TOC content. They indicate that these clusters likely correspond to nannofossil assemblages thriving during intense upwelling. Conversely, lower productivity and, thus, higher water column oxygenation are marked by a correspondence of clusters 2 and 4a with higher Mn/Al values, denoting a less intense oxygen minimum zone.

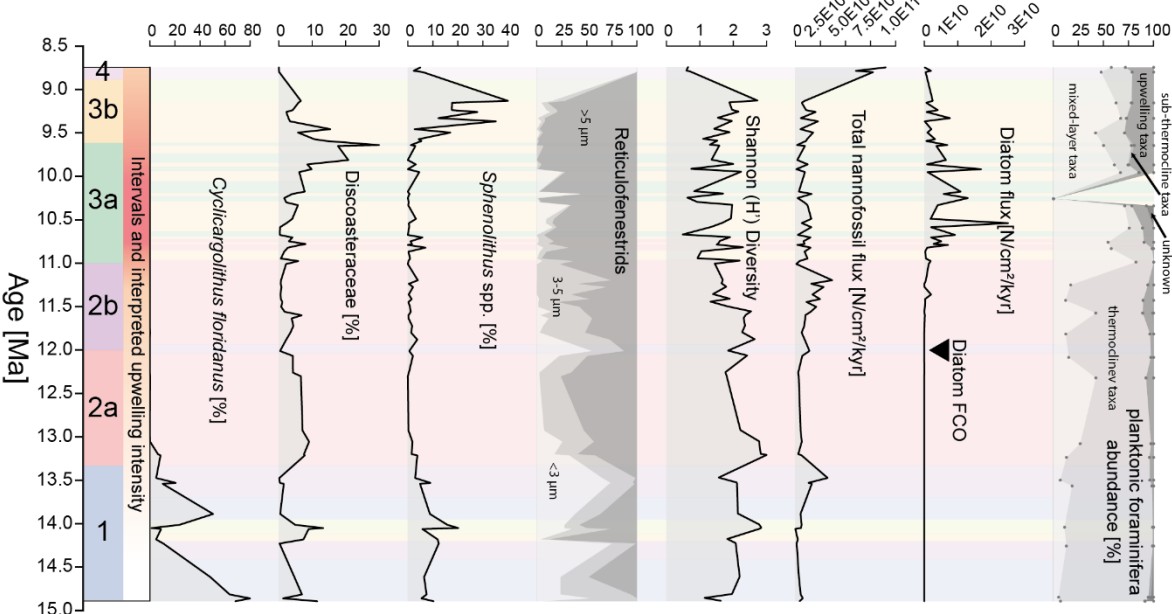

Figure 5: Summary of relevant nannofossil taxa (C. floridanus, the sum of all Discoasteraceae, *Sphenolithus* spp., as well as all 3 selected size ranges of *Reticulofenestra* spp.) shown as % abundance of the whole assemblage. Reticulofenestrids are combined into a single abundance graph showing the internal variability of the three defined size ranges of the genus Reticulofenestra. The Shannon (H') diversity is offered as an overall indicator of nannoplankton diversity throughout the study interval. The total abundance of nannofossils fluxes in N/cm²/kyr illustrates the stark increase in nannofossil accumulation in interval 4, denoting the noted bloom in small reticulofenestrids after 8.8 Ma. Next, the nannofossil abundances are contrasted with diatom fluxes. The nannofossil assemblage variability is further shown with classical upwelling indicators based on planktonic foraminifera, which shows an overall constant abundance of upwelling indicative taxa (e.g., *G. bulloides*) between Interval 3a and 4, despite the dynamic changes in the phytoplankton data. On the left of the figure we also show intervals 1 – 4 and their respective sub-intervals a/b and the resulting interpreted upwelling instensity. All data is unterpinned by the assigned clusters as defined in Figure 4.

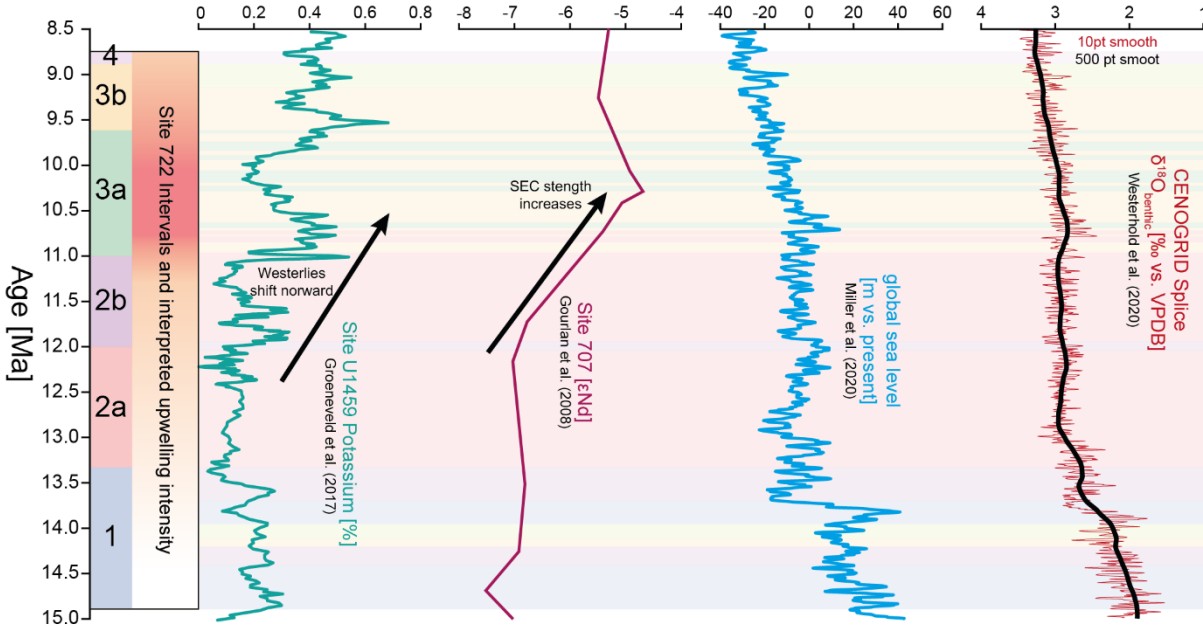

**Figure 6: Compilation of Indian Ocean and Global Data during the Study interval. Proposed plankton community**
**intervals as well as nannofossil assemblages at Site 772 are presented next to the abundance of natural gamma radiation**
**derived potassium content at Site U1459 (Groeneveld et al., 2017), interpreted to relate to precipitation changes in**
**western Australia as a consequence of the northward shifting southen hemisphre westerlies. The eNd data of Gourlan**
**et al. (2008), showing an increase in εNd signatures derived from Indonesia Indicating an increase in SEC strength, due**
**to a global increase in global ocean and atmospheric circulation (e.g., Betzler and Eberli, 2019). The global sea level**
**reconstruction of Miller et al. (2020)showing stable sea levels after the MMCT until at least 11 Ma. The global stable**
**CENOGRID stable oxygen isoptpe stack for the study interval, showing stable deep water conditions until 11 Ma**
**(Westerhold et al., 2020). On the left of the figure we also show intervals 1 – 4 and their respective sub-intervals a/b and**
**the resulting interpreted upwelling instensity. All data is unterpinned by the assigned clusters as defined in Figure 4.**

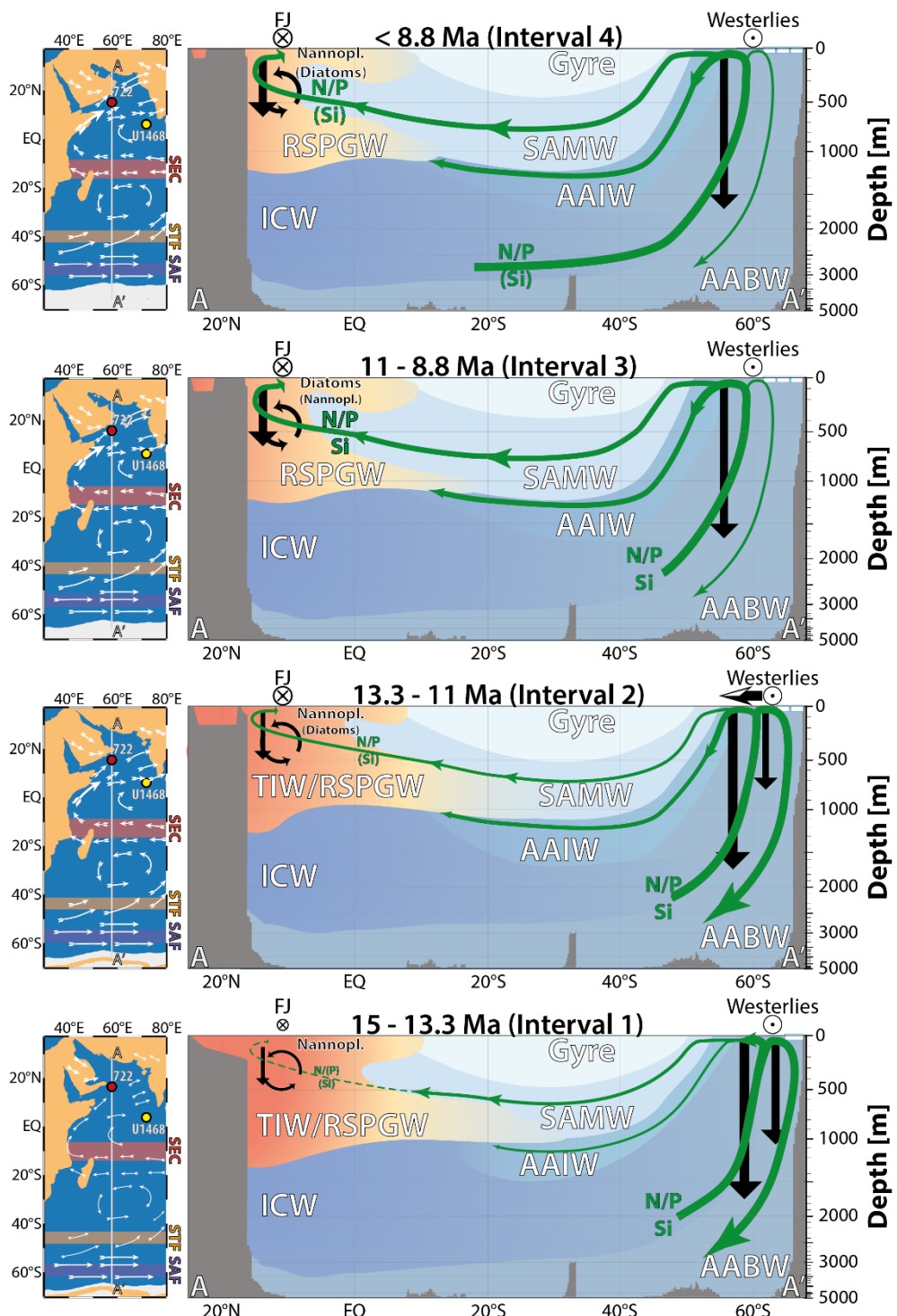


**Figure 7: Envisioned progression of upwelling along the Oman Margin based on paleogeography of Cao et al. (2017), adapted with regional information (Rögl, 1999; Bialik et al., 2019; Reuter et al., 2009, 2008), combined with hypothesized changing intermediate water-based nutrient supply throughout the study interval (c. 15 – 8.8 Ma). The figure also shows the hypothesized change in water masses over the study interval. Orange shading represents local water masses forming in the northern Indian Ocean migrating southward. While intermediate waters able to progressively migrate further in the the Arabian Sea where the begin to dominate upwelling by c. 11 Ma. Shading of the water masses represents their progressive intermixing with each other. Water masses shown are the Tethyan Intermediate Water (TIW), the Red Sea and Persian Gulf Intermediate Waters (RSPGW), Indian Central Water (ICW), southern Indian Ocean gyre waters (Gyre), sub-Antarctic mode water (SAMW), and the Antarctic intermediate water (AAIW) and Antarctic bottom water (AABW). In addition, note the corresponding change in nutrient (N, P, and Si) transport – vizualized by green arrows - following the proposed northward migration of the southern hemisphere westerlies due to sea ice expansion after 12 Ma (Groeneveld et al., 2017). Hypothesized changes in nutrient transport are based on model studies, which predict reduced low-latitude productivity during warmer climates (Laufkötter and**

**Gruber, 2018; Moore et al., 2018). Black arrows indicate the changes in the fluxes and hypothesized recycling of organic**
**matter within the WAS upwelling zone.**
**Table 1: Ecological interpretation of the defined nannofossil taphogroups based on the ecological parameters of the**
**defining nannofossil taxa.**

| Tapho-group | Defining Taxa | Ecology | References | Environmental Parameters |
|---|---|---|---|---|
| TG1a | *Reticulofenestra minuta* dominant | Dominated by r-selected opportunistic nannofossil taxa. Commonly interpreted as nutrient elevation in the photic zone. | (Haq, 1980; Wade and Bown, 2006; Auer et al., 2015) | Associated with high calcium carbonate accumulation |
| TG1b | Small and medium reticulfenetrids together with *Cyclicargolithus floridanus* | Warm to temperate waters, with increased nutrient conditions. | (Wei and Wise, 1990; Wade and Bown, 2006; Auer et al., 2015) | Associated with high calcium carbonate accumulation |
| TG2 | *Cyclicargolithus floridanus* and common medium reticulofenestrids | Warm to temperate waters, with moderate nutrient conditions. | (Wei and Wise, 1990; Wade and Bown, 2006; Auer et al., 2015) | Associated with high Mn/Al ratios (= weak OMZ) and elevated carbonate content |
| TG3 | Large reticulofenetrids dominant with common Discoastrids | Elevated nutrient conditions with deep nutricline and possible (seasonal) stratification | (Lohmann and Carlson, 1981; Backman et al., 2013; Imai et al., 2015, 2017) | Associated with biogenic silica, TOC, dust flux and lowered Mn/Al ratios (=stronger OMZ) |
| TG4a | Variable small, medium and large reticulofenestrids with common *Sphenolithus* spp. and discoasterids | Elevated nutrient conditions with high seasonal variability and intermittent stratification, possible indication of increased environmental stress. | (Castradori, 1998; Blanc-Valleron et al., 2002; Gibbs et al., 2004b; Wade and Bown, 2006; Villa et al., 2008; Beltran et al., 2014; Imai et al., 2015; Schueth and Bralower, 2015) | Weakly associated with carbonate accumulation and higher Mn/Al ratios (= weak OMZ) |
| TG4b | Large reticulofenestrids dominant | High nutrient conditions, likely open marine and potentially stratified. | (Auer et al., 2014, 2015; Beltran et al., 2014; Imai et al., 2017, 2015) | Weakly associated with biogenic silica flux, TOC and reduced Mn/Al ratios (= increasing OMZ) |
| TG4c | Medium and large reticulofenestrids dominant | High nutrient levels, likely upwelling derived. | (Haq and Lohmann, 1976; Lohmann and Carlson, 1981; Wade and Bown, 2006; Auer et al., 2014, 2019) | Not associed with Mn/Al ratios (= strong OMZ), no strong association with other paramters |


**Table 2: Interpretation of habitat depth of the identified planktonic foraminifera taxa.**

| *Taxa* | *Habitat* | *Reference* | *Comments* |
|---|---|---|---|
| *Dentoglobigerina altispira* | **open ocean mixed-layer** | (Berggren et al., 1985; Aze et al., 2011) | Symbiont bearing |
| *Fohsella fohsi* | **open ocean thermocline** | (Aze et al., 2011) | |
| *Fohsella peripheroronda* | **open ocean thermocline** | (Berggren et al., 1985; Aze et al., 2011) | Extends to cool subtropical waters |
| *Globigerina bulloides* | **upwelling** | (Kroon et al., 1991) | |
| *Globigerina* sp. | **open ocean mixed-layer** | (Aze et al., 2011) | |
| *Globigerinita glutinata* | **open ocean mixed-layer** | (Majewski, 2003; Pearson and Wade, 2009) | |
| *Globigerinoides obliquus* | **open ocean mixed-layer** | (Nikolaev et al., 1998) | |
| *Globigerinoides ruber* | **open ocean mixed-layer** | (Nikolaev et al., 1998) | Symbiont bearing |
| *Globigerinoides* sp. | **open ocean mixed-layer** | | Based on another present taxa of this genus |
| *Globoquadrina dehiscens* | **open ocean thermocline** | (Pearson and Shackleton, 1995; Nikolaev et al., 1998) | Noted to be erratic and variable by Pearson and Shackleton (1995). |
| *Globorotalia archaeomenardii* | **open ocean thermocline** | | Based on similarities to *G. manardii* |
| *Globorotalia menardii* | **open ocean thermocline** | (Regenberg et al., 2010) | |
| *Globorotalia plesiotumida* | **open ocean thermocline** | (Aze et al., 2011) | |
| *Globorotalia scitula* | **open ocean sub-thermocline** | (Itou et al., 2001) | *G. scitula* flux is inverse to POC flux |
| *Globorotalia* sp. | **open ocean thermocline** | | Based on another present taxa of this genus |
| *Globorotaloides hexagonus* | **upwelling** | (Spezzaferri, 1995) | May also be deep sub-thermocline dweller (Brummer and Kučera, 2022) |
| *Globoturborotalita druryi* | **open ocean mixed-layer** | (Kennett and Srinivasan, 1983; Aze et al., 2011) | Symbiont bearing |
| *Globoturborotalita nepenthes* | **open ocean mixed-layer** | (Aze et al., 2011) | |
| *Neogloboquadrina acostaensis* | **open ocean thermocline** | (Aze et al., 2011) | |
| *Orbulina universa* | **open ocean mixed-layer** | (Aze et al., 2011) | |
| *Paragloborotalia mayeri* | **open ocean thermocline** | (Aze et al., 2011) | |
| *Sphaeroidinellopsis seminulina* | **open ocean thermocline** | (Aze et al., 2011) | |
| *Sphaeroidinellopsis* sp. | **open ocean thermocline** | (Aze et al., 2011) | |
| *Trilobatus quadrilobatus* | **open ocean mixed-layer** | (Chaisson and Ravelo, 1997) | Deep mixed layer in Nikolaev et al. (1998) |
| *Trilobatus sacculifer* | **open ocean mixed-layer** | (Aze et al., 2011) | Symbiont bearing |
| *Trilobatus trilobus* | **open ocean mixed-layer** | (Aze et al., 2011) | Symbiont bearing |