# Peer review of "Biotic Response of Plankton Communities to Middle to Late"

_Climate of the Past, 2023_

## Author Comment (AC1)

**Reponse to anonymous Referee #1**

We would like to thank referee #1 for their thorough and very constructive comments. Below follows a point-by-point response to the comments and changes that will be enacted in the revised manuscript. Our answers are indented for differentiation from the original text by referee #1.

**General comments**

The present contribution traces the evolution of the upwelling system in the NW Indian Ocean (and its atmospheric and oceanic drivers) over the time interval from 15 to 9 million years ago (mid- Miocene) and is based on data raised from sediment cores of ODP Site 722 on Owen Ridge. Counts of calcareous nannofossil taxa, siliceous fragments and planktonic foraminifers are presented in context with data on sedimentology, geochemistry and isotopic composition of the sediment sequence. The new microfossil assemblage data are interpreted as indicators of nutrient conditions in the mixed layer and in the thermocline, and the record is proposed to illustrate interactions between different plankton groups and varying nutrient levels and ratios. The data and interpretations of local developments are then discussed in perspective of regional and global changes in the coupled ocean-atmosphere system, which at that time experienced major reorganization in response to tectonic processes and opening/closing of ocean gateways.

Novelty of the manuscript is in the statistical investigation of quantitative nannofossil data in 71 samples (for nannofossils and siliceous debris) and planktonic foraminifers in 28 samples and their implications for reconstruction of trophic state, nutrient limitations, and paleoproductivity in the mixed ocean surface and thermocline. The statistical analysis of these data yields four "taphogroups" (plus subgroups) that are proposed to be indicative of specific local surface water conditions, mainly of nutrient concentrations and nutrient ratios, and are employed to track the evolution of conditions at Site 722 through time. This reviewer is not qualified to evaluate whether the approach of using size- and morphology-based traits as expressions of nutrient regimes is standard practise, particularly in light of the stated "weak support for individual clusters reflecting the overall strong similarities in the assemblage composition of the studied samples" (line 212). The abstract of the Paasche (2010) publication on "Roles of nitrogen and phosphorus in coccolith formation in Emiliania huxleyi" suggests that reactions of this particular modern prymnesiophyte to nutrient limitations are much less straightforward than stated here.

> **Reply:** We thank the reviewer for this comment, as it clarifies that the MS needs to be more precise how the used techniques and proxies result in our given interpretation. In the revised version, we will be refining these sections accordingly.

Ancillary proxies, such as XRF data (dust and Mn), carbonate-C and organic-C, delta15N, have previously been published in Bialik et al. ( 2020). Why these are suitable proxies is unclear, because the introduction is incomplete and sections of text appear to be out of sequence and/or truncated in the presentation of proxies used to track wind, upwelling and OMZ (lines 93-101). In many cases the authors use these ancillary proxies to bolster their arguments based on the nannofossil assemblage, which occasionally results in circular reasoning.

> **Reply:** Similar to the above comment, this albeit broad critique can be seen as valuable information regarding missing clarity in the text. These proxies are commonly applied for said interpretation and follow the same guidelines as already outlined in Bialik et al. (2020). It was our oversight to assume that the validity of these proxies is universally accepted and, following published works, is sufficient to provide a working hypothesis for the further interpretation of paleobiological data in the context of these proxies.

Therefore we will be revising the introduction accordingly, aiming to explain the proxy validity better and fully justify the need for their application in this study.

As a main conclusion, the authors propose that the atmospheric (SW monsoon) driver and an incipient oxygen minimum zone existed in the western Arabian Sea, but that a one-million-year lag in productivity and community response to upwelling marks a teleconnection to southern hemisphere intermediate water formation: Only after intermediate waters formed in the Antarctic Convergence imported new nutrients into the thermocline of the northern Arabian Sea did the upwelling elicit a typical high-productivity response in the plankton community. Furthermore, they propose that changing nutrient ratios (caused by variable denitrification in a variable oxygen minimum zone) subsequently dictated changing patterns of planktonic ecosystems at continued high nutrient supply.

The manuscript thus offers interesting observational evidence of previously published modeling results and is an extension of a previous publication of some of the authors. The positive view is dampened somewhat by the (frequently unreflected, sometimes counterfactual and occasionally contradictory) rendition or reiteration of statements that are at least worthy of discussion. This particularly concerns the interactions between atmospheric drivers, water mass distributions and their nutrients, and dynamics of the oxygen minimum zone in the various subchapters of the discussion and the discussion of global implications. These parts of the manuscript definitely need careful scrutiny.

> **Reply:** While the assessment that our rendition is „frequently unreflected, sometimes counterfactual and occasionally contradictory" seems somewhat counterproductive as it offers no constructive criticism, we will nevertheless take this critique seriously and will follow the detailed comments and questions closely to alleviate this perceived issue raised reviewer#1.

The manuscript fits the scope of the journal, in particular in a special issue dedicated to Dick Kroon, who has laid grounds for the reconstruction of the monsoonal upwelling in the Arabian Sea. It is reasonably well written, but comparatively long for what the authors have to say, and the list of references is very long indeed. In a revision, text should be edited with care and preferably by a native speaker. Some passages of text on intricacies of placolith morphologies should be omitted. The continual use of comparatives without reference to what it is compared to must be corrected. Figures are ok, but may possibly be condensed by plotting factor scores instead of single species´ abundances.

> **Reply:** This paragraph contains several essential aspects which we would like to comment on:
>
> 1.) We take note of the comment on the length of the MS and will shorten the revised version according to the suggestions of the reviewer as outlined in the detailed comments.
> 2.) Why the reviewer seems to see a thorough list of pertinent citations as a negative is difficult to understand, especially as they note additional works in their review that we have not cited.
> 3.) Why „intricacies" of the applied taxonomy should be omitted is beyond our understanding, as they are essential to explain our data basis. Irrespective of the taxonomic importance, it furthermore, is also a necessary aspect for the following interpretation and, indeed, for understanding our interpretation in general.
> 4.) Plotting factor scores is an interesting suggestion. However, it would remove the visualization too far from the original data, which we would like to avoid as much as possible. Hence, we decided to show the original data, as was already done in our original submission.

**Detailed comments and questions:**

Many of the data and many of the arguments in the discussion echo the Bialik et al. (2020; Paleoceanography and Paleoclimatology) publication: It is necessary to highlight the novelty of the present paper more clearly and succinctly.

> **Reply:** We thank the reviewer for pointing this out. The novelty, is that we, for the first time, can show the biotic response of primary producers to these changes and propose a potential mechanism to rectify the diachronous history of proxy data in the existing literature. We are sorry that this was not made clear enough for the reviewer in the current MS, and we will revise the MS with this comment in mind.

A number of conclusions in that publication are taken for granted here, so that some of the questions below are addressed to the precursor publication. The following more or less relevant points came to my mind when reading the paper:

- The observed patterns are attributed to nutrient import from sub-thermocline waters originating in the Antarctic Confluence. What about changes in Indonesian Throughflow that progressively limited water exchange and initiated formation of the Indian Ocean Central Water mass that is the principal source of upwelling water offshore Oman (e.g., Kuhnt et al., 2004; You and Tomczak, 1993; You, 1997)? Not all readers may know what water masses are involved, so that a brief rendition of circulation the modern Indian Ocean that was established near the end of the investigated time interval is in order.

  > **Reply:** It is essential to also consider available paleogeographic reconstructions of the ITF region (see Hall, 2012; DOI: 10.1017/cbo9780511735882.005), which by no means are as clear on the restriction history of the ITF region between 15 and 8 Ma as the reviewer suggests at this point. According to these paleogeographic reconstructions of the region, it is likely that at least until 10 Ma, there was no deep connection between the Pacific and Indian Ocean through the ITF region. Likely the deep trenches were not yet present, and shallow island and carbonate platforms proliferated between Australia and Sundaland. These reefs, therefore, likely largely restricted intermediate and deep water connection through the Indonesian Archipelago during that timeframe (see Figures 3.10 and 3.11 in Hall, 2012). You (1998) further notes that transformed AAIW may rise up into the thermocline layer in the Arabian Sea. We fully agree that this may not be general knowledge and thus have included a more detailed description as well as a new panel in fig 1 to better illustrate these aspects of our interpretation.

- What about the role of uplift and changes in elevation in central Asia as a driver for the SW (and NE) monsoon inception? The text only refers to latitudinal temperature gradients and emergence of the Arabian Peninsula.

  > **Reply:** Please refer to Sarr et al., (2022; DOI: 0.1038/s41561-022-00919-0), for details. This has been modelled and very clearly shown in previous works and should not require further discussion. To quote from their work [***inline citations omitted for brevity***]: „*This emerging view does not preclude an important role for the Himalaya–Tibet orogeny in increasing rainfall amounts and in the establishment of a longer rainy*

*season (similar to modern) in the early Miocene or before. However, our results do indicate that tectonic activity in the Himalayan and Tibetan Plateau regions cannot account for either the palaeoceanographic changes observed in the Arabian Sea and equatorial Indian Ocean or the establishment of modern-like large-scale atmospheric circulation with a strong Somali Jet"*

As this was apparently not stated clearly enough in the introduction, we changes this section of our manuscript to the folowing:

- Is upwelling above Owen Ridge indeed driven by the Findlater Jet, or by wind stress curl? How does emergence of the Arabian Pensinsula (which is discussed as a decisive factor for unclear reasons) influence that?

  **Reply:** This is quite easily answered: We follow the most recent model-based assessment of the system provided by Sarr et al. (2022). To quote from their abstract, *"The uplift of the East African and Middle Eastern topography played a pivotal role in the establishment of the modern Somali Jet structure above the western Indian Ocean, while strong upwelling initiated as a direct consequence of the emergence of the Arabian Peninsula and the onset of modern-like atmospheric circulation. Our results emphasize that although elevated rainfall seasonality was probably a persistent feature since the India–Asia collision in the Paleogene, modern-like monsoonal atmospheric circulation only emerged in the late Neogene"*. We have clarified this in there revised manuscript.

- Seasonality is an important influence on plankton succession and alternating eutrophic/oligotrophic deposition at the transition from coastal to open-ocean upwelling (such as is the case over Owen Ridge). Is it possible that the entire phytoplankton assemblage between 13 and 10 mya reflects extreme, but variable seasonality, not just at times of a dominant taphogroup 3?

  **Reply:** This is indeed a good point. However, the most notable difference to other taphogroups is the occurrence of comparatively (to other taphogroups) high amounts of Discoasterids, which are regarded as generally oligotrophic, and are only observed in Taophogroup 3. This would indicate that this taphogroups exhibits high (likely Summer Monsoon Drivel) upwelling and productivity, leading to increased productivity that can consume nutrients rapidly. In non-upwelling months, this high productivity thus results in nearly oligotrophic surface water conditions. This pattern is only observed in Taphogroup 3, hence our description as extreme seasonality. So to conclude, seasonality is likely always present. However, the noticeable co-occurrence of high productivity indicators and generally oligotrophic indicating microfossil taxa is unique to TG3.

- How did an OMZ form at low productivity and C-flux around 13 Ma?

  **Reply:** Low productivity and C flux at Site 722B do not preclude a more limited upwelling cell closer to the coast, which has provided sufficient OM flux to establish a limited OMZ in the region.

- Figure 2 and 3 plot abundances of individual taxa that are representative of specific conditions. Would a representation (similar to Fig. 5) of scores for the clusters (or factors) in Figure 4 show a clear pattern of changing conditions in mixed-layer plankton, monsoon strength and nutrients?

**Reply:** We understand why the reviewer made this suggestion. However, our firm opinion is that showing primary data has more value for assessing assemblage variability. Hiding gathered data behind derived statistical output may obscure clear patterns that are also visible in the observable data.

- Why not calculate accumulation rates of TOC and opal instead of concentrations (often used in the text to indicate accumulation, which is not correct due to the role of dilution) from the age model and GRAPE values?

    **Reply:** An excellent suggestion, although GRAPE data only provide wet bulk densities, which is not ideal for defining fluxes, as water content (and thus porosity) would introduce a large bias on mass accumulation rate calculation. We have therefore used MAD dry bulk density to calculate fluxed for all relevant parameters (including individual nannofossil taxa, as $N/cm^2/kyr$), but ultimately decided against using it in this publication. This decision was based on the fact that the age model and the dry bulk density are of relatively low resolution and may have introduced additional errors.

    Following the comment of referee#1, which we agree with, we have nevertheless re-run all relevant statistical analyses with flux data and have found no significant difference in the results compared to the ones presented. Hence we have, after some deliberation and weighing the pros and cons of this approach, decided to change all relevant figures and discussions to refer to fluxes in the revised version of the MS.

- Delta15N values from Site 722 appear to be significantly lower than late Pleistocene values. What is the reason? Apparently, the data were raised on acidified samples (according to methods section in Bialik et al., 2020), which introduces spurious results. A "denitrification threshold" at 6 permil is not likely, when thermocline nitrate originating in the AA confluence has a value of more than 5 permil then and now.

    **Reply:** Bialik et al., (2020) follow the approach of Tripathi et al. (2017), which we also apply herein. We note that even the data or Tripathi never reaches $\delta^{15}N$ values above 8‰ in the Pleistocene. To quote from Tripathi et al. (2017) [***inline citation omitted for brevity***]: „*Based on surface sediment analysis of more than 100 locations in the Central and Eastern Arabian Sea (most of them are located in the Eastern Arabian Sea), the $\delta^{15}N$ values of SOM have been found to vary from 6‰ to 11‰. In most of the oxygenated basins, the $\delta^{15}N$ values do not exceed 6‰ while those from the oxygen deficient basins are highly enriched with mostly higher than 6‰. Thus, the periods with $\delta^{15}N$ values higher than 6‰ may signify denitrification associated with strong OMZ.*"* This explanation will be included in the revised manuscript as it was clearly missing from the present version leading to a lot of confusion on the interpretive basis, which we have applied to our proxy data.

- The origin of coastlines used in Figure 6 is unclear and they do not really illustrate the prominent role of emergence of the Arabian Peninsula proposed in the text. In fact, the panels all look pretty similar to me (except the hand-drawn lines supposedly illustrating nutrient import): what is the line and label N:P/Si supposed to mean? As far as I know, SAMW is not the upwelling water mass today as suggested in the figure, but that should possibly be treated somewhere in the text (for modern, aka post-Miocene conditions).

    **Reply:** Please see the relevant citations supplied with the manuscript to justify this assumption. However, we agree that this needs to be made more evident in the MS. Hence we also intend to add an additional panel to Figure 1 to illustrate the presentday expansion of SAMW above the equator in the Indian Ocean and its mixing path with Indian Central Water and the Arabian Sea and Persian Gulf high salinity waters. We consequently see no reason why the same model should not be assumed as true in the past when we have clear evidence similar to the present-day paleogeographic configuration in the Indian Ocean.

Regarding the paleogeographic reconstructions presented: These are based on paleogeographic maps of Cao et al. (2017) and several works detailing regional information on the uplift of the Arabian Peninsula and potential marine sedimentation during the Miocene there. This information was erroneously lost in the figure caption in the previous version and is now remedied. We apologize for the confusion and thank the reviewer for catching this oversight.

**Detailed comments keyed to line numbers:**

I find the statement in first sentence of the abstract difficult to understand. Why is that so?

**Reply:** Suggesntions like these are difficult to follow as the offer no concrete evidence as the what caused the difficulty in understanding our texrt. We have attempted to revise the sentence noted for clarity

In my understanding, upwelling cells are localized spots of high upwelling intensity caused by the interactions of wind and local topography. Many of these cells combined form upwelling systems. One is the WAS upwelling system, the only major western boundary upwelling system.

**Reply:** We agree and apologize if the MS text did not make this clear enough in its present form.

Concentration of TOC in the sediment says nothing about accumulation! Take care to not use these two terms synonymously….

**Reply:** While not the same, these two are still intrinsically related. The comment of REV#1 pre-supposes a clear shift in sediment accumulation which is not present in the studied interval based on existing age model information. We have nevertheless decided to use MAR calculations in the revised version of the MS, as we agree it is a good suggestion.

Upwelling acts as both sink and source of CO2 to the atmosphere, and each system differs in the net balance.

**Reply:** Yes. We have clairifed our MS.

does this mean number of nannos per g of CaCO3 or divided by CaCO3? Unusual annotation! Are two digits after the decimal within the confidence limits of your method?

**Reply:** Unusual, true, but we felt it was rather elegant to show changes of total abundance irrespective of „dilution of the carbonate flux" by biogenic silica. We are sorry that the reviewer disagrees with this assessment and therefore have chosen to use MARs instead.

231ff So, are the results statistically robust, or not?

**Reply:** They are, it was just necessary to discuss why the stress was above <0.1 in the nMDS. As the uncertainly of the reviewer likely stems from our poor writing style we have revised this portion to hopefully clarify this point.

…2, whereas

**Reply:** Unclear what the reviewer requests of us here.

placoliths are the small plates, arent´t they? How can they proliferate?

**Reply:** This seems to be a misunderstanding based on semantics. We we will revised this portion fo the MS for clarity. Briefly: An increase in small placoliths corresponds to an increase (and therefore proliferation) in small placolith-producing coccolithophores.

263-266 How can a "highly productive open marine environment" be nitrogen limited? This means an excess of phosphate and at that point, nitrogen fixation should kick in to make up the deficit. Please explain.

**Reply:** Please see Pearl (2018), for a brief summary on this issue. We have clarified the text in the MS accordingly.

elevated sources?

**Reply:** Cryptic statement. We assume it pertains to the interpretation of nutrient requirements of R. minuta. This is a valid assumption, but such a source shift would also be present in the basic sedimentology of Site 722, which was already evaluated by Bialik et al. (2020). No shift terrigenous sources were detected that could explain this shift in assemblage patterns.

N-limited nutrient sources meaning low N:P ratios in upwelling thermocline water?

**Reply:** Thank you for pointing out our lack of clarity, which led to this statement. Not necessarily. Please refer to Paerl (2018) for the biochemical basis of this statement. Briefly: N-limitation often persists in environments where denitrification and upwelling take place. This lack of bioavailable N is caused by environmental factors such as high $O_2$ concentrations in the mixed layer and turbulence, among others – leading to a suppression of N-fixation and, therefore, N-limitation. As the author describes it, this process often leads to „chronic N limitation", even in the present-day ocean, where anthropogenic nitrogen enrichment is present. We intend to clarify this portion of the MS in the revised version.

elevated nutrient levels compared to what? Frequently the comparative is used throughout the text without reference to what the comparison refers to.

**Reply:** Thank you for pointing this out. Have clarified this statement in the following way: assemblage indicates elevated nutrient sources levels, compared to a fully oligotrophic assemblage

277-278 The thermocline and nutricline coincide usually and there the nutrient levels are high at the base – that is why the plants are there in the first place even at low light levels (deep chlorophyll maximum) …..so, what does "elevated nutrient conditions" mean?

**Reply:** We have revised this section for clarity.

How does this setting differ from conditions of TG 1(b)?

**Reply:** In the way it does show a clearly different fossil composition to TG1b, and is interpreted differently. No changes, apart from minor additions for clarity, have been made.

concentrations, not accumulation rates!

**Reply:** Thank you for pointing this out. We have now included calculated MARs. However, The patterns are very much unaffected due to the overall linear sedimentation rate and stark changes in accumulation/MAR increases.

Why should that high dust flux increase productivity? Or is that a consequence of enhanced ballasting and export flux?

**Reply:** Dust flux, in this case, is both a rough indicator of an increase in the strenght of the Findlater/Somali Jet, and may indeed also supply Fe as a micronutrient to settings/intervals where both N- and P-limitation are no longer a factor. This has been clarified within oure revised MS.

more abundant than what?

**Reply:** This was revised.

How could such a nitrogen-excess situation arise? Very difficult to imagine at fixed N:P ratios in thermocline waters and active denitrification in the OMZ! It appears that you put a lot of trust in a limited set of culture data for recent N-cell clones of *Emiliania huxleyi*.

**Reply:** Unclear what the reviewer intends with a comment worded in this fashion. We tried to revise for clarity.

strong upwelling

**Reply:** Again unclear what this comment intends. Revisions for clarity were attempted.

299-301 weaker, stronger, higher than what?

**Reply:** Revisions for clarity have been made.

P-limitation (see query above)

**Reply:** See response above.

what is an active OMZ?

**Reply:** We revised for clarity.

(and 344) more limited than what? If there is no upwelling and only low productivity, how would an OMZ form in the first place? What is the exact link between monsoon strength and the OMZ without an intermediate link created by organic matter rain rate and water mass residence time?

**Reply:** The reviewer seems to intentionally misread our text here. As we assume this to be their way of showing concerns in terms of readability for a more general readership, we made every effort in clarifying this section.

How and why is intensified upwelling linked to high delta15N?

**Reply:** See above.

what amplifies? Declining upwelling?

**Reply:** Revised.

I may have missed an explanation, how upwelling intensity is recorded in delta15N values.

**Reply:** See above.

What is the role of the temperature gradient today? Is it the influence on the sea level pressure gradient? How is the deep-water temperature gradient (between which end members?) involved?

**Reply:** Unclear comment. The statement is explained and cited on lines 415-417. This includes a citation (Gadgil, 2018) which covers the modern basis of the summer monsoonal forcing.

gradients, thereby

**Reply:** Thank you, fixed.

indicated by instead of related to?

**Reply:** We did not follow this suggestion, as it would change the intended meaning. We revised the sentence for clarity.

I must have missed the link between Mn/Al ratios and productivity. OMZ intensity comes to mind, but how is that linked to low productivity in a low nutrient regime?

**Reply:** We appologize if this was not made clear, please see our revised introduction and mehtods section.

As a result of? Explain the link between SST, sea level and upwelling!

**Reply:** We have revised the sentence as follows: *Therefore, we link our new assemblage data with an extensive data compilation highlighting a progressive upwelling increase, which leads to thermolcine shoaling. This thermocline shoaling in turn results in declining sea surface temperatures and increased surface water productivity through the upwelling nutrient rich thermocline waters along the Oman Margin during this time (Fig. 3; Zhuang et al., 2017; Bialik et al., 2020a).*

why poorly ventilated? In line 461 below you state that that increase/decrease is indication of a shallow and poorly vented thermocline - what changed?

**Reply:** This is a statement of facts, hence the citation. An interpretation follows thereafter.

this is not accumulation! You might have a lot of TOC raining down, but when it is diluted by a lot of dust, for example, TOC concentrations are low!

> **Reply:** See comments above. We agree this was written somewhat imprecisely and we will revise the MS accordingly.

formation of nutrients? Are they formed, or are they not used up because of light limitation?

> **Reply:** Changed to „retention"

mineralizing primary producers – is that a commonly used term? Other people use that word for dissimilation of organic matter and nutrient release

> **Reply:** Mineralizing is used in the sense of primary producers which are „capable of forming a mineral compound" in this context. Similar to how „calcifying" would denote calcite forming primary producers. We chose to be more specific here, as we have no evidence on the total biomass of cyanobacteria and other non-mineralizing (hence the use of this term) primary producers.

505-514 This entire discussion is very difficult to follow and possibly not suitable here: You infer from a size shift in one genus that the nutrient regime changed, but then discount this explanation and invoke changing nutrient limitation, but do not state the nature of that limitation.

> **Reply:** The reviewer may to have misread the statement made in the context the discussion chapter as a whole. Changes where made to the text for clarity nevertheless.

– 520 the concept of Mn-redirection was lost in the introduction. Do you talk about sediments, or water? Are high concentrations in sediments seen at the top and bottom of the OMZ where it intercepts the margin?

> **Reply:** We included a paragraph in the introduction.

Shifts in nutrient saturation? I don´t think that you can saturate seawater in nutrients.

> **Reply:** It was changed to content.

are you talking about N:P:Si ratios?

> **Reply:** Unclear statement by the reviewer. No revisions for clarity were found necessary after several passes by the authors.

intermixing with

> **Reply:** done

There is abundant literature on iron supply from continental margin sediments, particularly when they are situated in an OMZ

> **Reply:** We assume this is a request to add more citations at this junction, which we follow.

indicate a change in

**Reply:** done quantity of nutrient enrichment?

**Reply:** Revised.

Explain how that affects the northern AS (see above)! I am not entirely convinced that the record from the AS is compelling evidence….there may be other factors at play.

**Reply:** We recommend the following literature to alleviate these concerns: Laufkötter and Gruber (2018); Toggweiler et al. (2019); Böning and Bard (2009) and Taucher et al. (2022). Although we are not trying to convince the reviewer, we are here to present scientific evidence and discuss said evidence in the context of available literature. If the reviewer has further suggestions of literature we may have missed, they should have provided them! We would have been happy to discuss them in the context of our data.

what is "nutrient rejuvenation"?

**Reply:** Replenishment of nutrients from a source outside compared to local recycling of nutrients.

I am not sure I understand the argument for an increasing wind regime.

**Reply:** Revised for clarity.

Explain how and why wind shear increases, then causes a global shift in ocean-atmosphere circulation, and deepens the thermocline. In my view, increasing wind shear causes open ocean upwelling and shallowing of the thermocline!

**Reply:** These concepts have been covered in the introduction, and we certainly don't feel it is necessary to iterate such basic concepts at this stage.

nutrient poor. But how then do you explain the OMZ that is apparently evident at that time?

**Reply:** Done. Just because Site 722 exhibits lower nutrient levels, nothing precludes upwelling closer to the coast. We recommend to read these statements in their intended context.

became

**Reply:** Thank you.

624-626 According to You and Tomczak, 1993 and You, 1997, the upwelling taps essentially Northern Indian Ocean Central Water mixed with Red Sea/Persian Gulf waters.

**Reply:** We would have to strongly disagree with this somewhat outdated interpretation. In more recent studies, there is more than enough evidence that most upwelling north of the ACC is associated with conspicuous[14]C minima that match water signatures of pre-bomb of the SAMW. Thus most of the water upwelling in low latitude upwelling zones is supplied by mode or intermediate water from the ACC. See Toggweiler et al. (2019; DOI: 10.1029/2018JC014794). This can further be corroborated by a recent review of present-day water mass properties published by Böning and Bard (2009; DOI: 10.1016/j.gca.2009.08.028) for a more up-to-date understanding of how the subsurface waters in the WAS are predominantly ventilated by waters derived from Subantarctic Mode and Antarctic Intermediate Waters. Finally, we would like to point out that You (1998) also traces AAIW up to a latitude of 5°N.

We have revised to make these aspects more clear and more prominent as they are indeed critical for the proper understanding of our interpretation.

That drop in SST is certainly not exclusively linked to a specific water mass, or is it? Not to enhanced upwelling?

**Reply:** This has been discussed throughout the MS.

what kind of shift? Excess phosphate? Less silicate?

**Reply:** Revised for clarity.

you never refer to fluxes, but to concentrations. Would it not be simple to use your age model to actually calculate component fluxes from 722 GRAPE data?

**Reply:** We would never use GRAPE data for such calculations. The available GRAPE data is inherently biased, as it only supplies wet bulk densities. The better approach would be to use the discrete moisture and density (MAD) dry bulk densities (DBD) available for the Site. This is what we also intend to include in the revised manuscript, as we agree with the reviewer's concerns that this may result in a certain lack of clarity in this manuscript. Granted, there are approaches to using MAD-based DBD to correct GRAPE values. This approach, however, will introduce another potentially error-prone derivate into the calculations, as at the higher depths, compaction also has a large effect on dewatering pore spaces and, thus wet bulk density. Hence we selected a linear interpolation of the available MAD data.

delta13C is not shown in Fig. 3

**Reply:** No, but TOC is shown. We simply placed the figure reference at the end of the sentence. Revised in our current MS.

which environmental stressors aside from nutrients?

**Reply:** Revised for clarity

References not cited in the manuscript:

You, Y. and Tomczak, M, 1993. Thermocline circulation and ventilation in the Indian Ocean derived from water mass analysis. DSR I, 40-1, 13-46.

You, Y., 1997. Seasonal variations of thermocline circulation and ventilation in the Indian Ocean. JGR, 102/C5, 10391-10422.

**Reply:** Thank you for the suggestions. After some consideration we have elected to include these citations together with You (1998). We however note that You (1998) represents a more recent evaluation of intermediate water masses in the Indian Ocean by You and colleagues, which represents a re-evaluation of several „history" Indian Ocean dataset after the World Ocean Circulation Experiment (WOCE; see You, 1998).

---

## Author Comment (AC2)

**Response to anonymous referee #2**

We would like to thank referee #2 for their thorough and very constructive comments. Below follows a point-by-point response to the comments and changes that will be enacted in the revised manuscript. Our answers are indented for differentiation from the original text by referee #2.

In the manuscript entitled "Biotic Response of Plankton Communities to Middle to Late Miocene Monsoon Wind and Nutrient Flux Changes in the Oman Margin Upwelling Zone" submitted to "Climate of the Past", G. Auer and co-authors retrace the evolution of the upwelling cell in the Western Arabian Sea over the Middle to Late Miocene interval ~15 to 8.5 Ma. The authors integrate counts of calcareous nannofossils and diatom frustules in 71 samples and counts of planktonic foraminifers in 28 samples (all converted to abundance %) with published geochemical data (XRF-scanning elemental data, carbonate, organic carbon and nitrogen isotope measurements) from Bialik et al. (2020). Statistical methods including cluster analysis are applied to investigate associations between microfossil abundances and environmental parameters derived from geochemical data.

The main findings of the research are: (1) the onset of upwelling at ~14 Ma along the Oman margin and development of full monsoonal conditions after ~13 Ma were closely linked to the evolution of regional tectonics and global climate; (2) a high-productivity regime was gradually established at ~12-11 Ma in tandem with high-latitude re-organization of intermediate-water formation (AAIW and SAMW); (3) peak upwelling productivity between ~12 and 9.6 Ma was driven by enhanced nutrient fluxes from increased AAIW and SAMW production; (4) upwelling productivity declined after ~9.6 Ma due to the waning of monsoonal winds.

The manuscript presents interesting, important results concerning a climate sensitive region within the Asian Monsoon system and it targets a period of the Middle to Late Miocene that has remained highly enigmatic. The multiproxy approach, combining records from calcareous and siliceous microfossil groups as well as geochemical data, provides valuable insight into the biotic response to long-term changes in upwelling-driven productivity along the Oman margin. The manuscript is well-suited to the scope of "Climate of the Past", in particular to an issue dedicated to Dick Kroon, who pioneered paleo-monsoon research in the Arabian Sea. Despite these positive aspects, I feel, however, that the manuscript requires some substantial revision. Please find below some major and minor issues that should be addressed before the manuscript can be considered for publication in "Climate of the Past".

> **Reply:** We thank reviewer for their thorough assessment of our manuscript. A detailed list of enacted changes an responses has been attached below. The original comments are presented for reference.

Major issues

This work extends previous research published by Bialik et al. (2020) in "Paleoceanography and Paleoclimatology". However, the amount of overlap between the two contributions is at times equivocal and will need to be clarified during revision. Clarification is needed, in particular, concerning the originality of the data and interpretations. Some reiteration of background information and reference to previous results from Bialik et al. (2020) appear in various parts of the manuscript, but relevant information is not always provided or easy to locate. For instance, Line 144 mentions that

portions of the nannofossil data set was already published, but does not indicate how many samples were previously analysed. Overall, the novelty of the findings in the current manuscript needs to be more clearly highlighted.

> **Reply:** Thank you for this recommendation, we have revised manuscript accordingly.

The manuscript is relatively long and I feel that the discussion on Lines 253-514 (Sections 5.1-5.3 of Discussion) could be streamlined to avoid some internal redundancy. In addition, the authors might consider summarizing the temporal progression of environmental changes outlined in Section 5.2 into a table for greater clarity. It would also be useful to include this information into one of the figures or into an additional summary figure. In the current version, the temporal evolution of upwelling productivity is not indicated on any of the figures and one has to search through the dense text to find this key information. The color coding in Figs. 2-3 and 5 only refers to the cluster assignment, based on the nannofossil assemblages.

> **Reply:** Thank you for this suggestion, we follow it closely during the revision of the manuscript. However definition of TG and Intervals should always be separate to avoid confusion about assemblage interpretation versus the application of changes in assemblage occurrence within a stratigraphic context. These are strongly separate aspects of interpreting the data, both of which should not be neglected or overlooked when working with fossil assemblage data, especially when applying multivariate statistics.
>
> We also agree that visualizing the complex aspects of the progression is an excellent suggestion. As a result, with have changed figure 3 and 5, to show the progression of interpreted upwelling intensity in conjunction with the intervals we defined based on our nannofossil taphocoenosis.

The manuscript contains abundant references to the recent literature to support the interpretation of the results and the discussion. To strengthen the interpretations and to highlight the key findings presented on Lines 515-691, it would be useful to add a synthesis figure that provides a direct comparison of results with some of the cited published records. At present, only the temperature data from Zhuang et al. (2017) are shown in Fig. 3. Adding a synthesis figure would be especially important to demonstrate, for instance, that major changes in global climate and ocean circulation are synchronous with changes in regional upwelling-driven productivity, as proposed in the discussion. Overall, such a figure would considerably help to bolster the interpretations.

> **Reply:** Thank you for this suggestion. We had hoped that our Figure 6 may serve this purpose, but it appears to have been less successful in doing so than anticipated. We have therefore included an additional figure (now figure 6), which summarizes a limited amount of relevant datasets in both the Indian Ocean region, but also two global compilations: a) the CENOGRID d18O data of Westerhold et al. (2020), and b) the sea level data of Miller et al. (2020). Figure 1 was also adjusted to show the Sites where the data in the Indian Ocean originates from.

The paper contains little information on present day circulation patterns in the Indian Ocean and on the origin of the water masses that upwell today in the Arabian Sea. As the authors put a great deal of emphasis on the role of distant circulation changes in controlling nutrient availability and ultimately productivity in the upwelling cell along the Oman margin, it would be useful to relate their reconstructions of past circulation in Fig. 6 to the modern scenario. I find the expansion of SAMW and AAIW to water depths of 1000 and 1500 m north of the equator during the Miocene somewhat surprising. However, I accept that very little is known about Miocene Indian Ocean circulation and that it may have markedly differed. Nevertheless, I still feel that other potential influences should be considered, such as the role of the Indonesian Throughflow, when the Indo-Pacific gateway was fully

open. I am also puzzled that the water depth of Site 722 is not taken into consideration (see comment below for Lines 121-124), when assessing the intensity of upwelling.

> **Reply:** This is a comment that has also been raised by referee#1. We note that there is clear modern-day evidence for the contribution of mode and intermediate waters forming in the ACC to low latitude upwelling cells (see Böning et al., 2009; Toggweiler et al., 2019; as well as Laufkötter und Gruber, 2018). As this is a critical aspect of our present study, we intend to add another panel to Figure 1 and be more specific in our introduction and discussion to give these aspects of our interpretation the necessary scientific basis.

Minor issues

Abstract

Lines 25-26: the sentence starting with "We combine…." is unclear. Please revise and clarify.

> **Reply:** Thank you, this has been revised.

Line 30: the duration of the MCO, based on distinct isotopic events, is considered to be 16.9 to 14.7 Ma. A global d18O decrease at 16.9 Ma signals the onset of the MCO and a global d18O increase at 14.7 Ma marks the first step in ice sheet expansion and global cooling during the MMCT (cf. Holbourn et al., 2014, 2015).

> **Reply:** This was revised as requested.

Lines 32-34: please specify time interval. Do you mean after 12 Ma?

> **Reply:** Thank you, we clarified that it happened after 12 Ma and persisted thereafter, at least based on our current data basis and interpretation.

Lines 35-36: please break up into two shorter sentences.

> **Reply:** Done, thank you for the suggestion.

Line 36: replace "beginning" by "the onset".

> **Reply:** Done, thank you.

Line 39: unclear what "SAM" refers to here. Acronym not previously explained.

> **Reply:** We apologize for this oversight, SAM is now spelled out at South Asian Monsoon. Thank you for pointing this out.

Line 40: "The absence of full correspondence…". Not really clear what is meant here.

> **Reply:** Has been revised to: „The absence of a clear correlation with (…)"

Line 43: omit "fossil" here.

> **Reply:** done

Introduction

Line 48: "a" missing before "biomass"

>   **Reply:** Thank you, we fix this typo.

Lines 54-60; upwelling areas can also be sources of CO2. Please revise.

>   **Reply:** While this is indeed is true on a short term basis, as CO2 degassing indeed releases ocean bound CO2 from the DIC system, we however have not come across any evidence that this short term release into the atmosphere can overcome the net CO2 sink based on productivity on geological timescales. Generally net C fluxes in upwelling zones remain skewed towards sequestration into the sediment (see Krapivin and Varotsos, 2016; DOI: [10.1016/j.jastp.2016.10.015](10.1016/j.jastp.2016.10.015)). We admit however that ocean atmospheric CO2 exchange should not be omitted in the introduction, hence why we have revised this section accordingly.

Line 68: on glacial-interglacial timescales rather than "in".

>   **Reply:** Revised.

Line 80: becomes established rather than "establishes.

>   **Reply:** changed

Line 93: sentence starting with "To date…" needs attention (seems incomplete).

>   **Reply:** Thank you, this sentence was indeed incomplete, which must have happened during editing of the final draft. We have now added the complete version of the sentence, which reads: To date, manganese redirection – i.e., the depletion of Mn in the sedimentary record due to Mn-reduction in the water column and subsequent advective transport to the edges of the OMZ – is one of the most used proxies to define OMZs and their past extent within the ocean (Dickens and Owen, 1994).

Line 102: see comment for Line 30 about duration of MCO.

>   **Reply:** Thank you, we added the more precise dates.

Line 104: verb should be plural (were established).

>   **Reply:** Thank you for catching this error. This portion, however, has changed due to revisions.

Lines 106-107: the MMCT usually refers to the Middle Miocene interval 14.7-13.8 Ma, and does not correspond to the Middle to Late Miocene, as implied here.

>   **Reply:** Thank you, we did not intend for this apparent imprecision. We only refert to cooling during the MMCT and subsequent upwelling intensification by 13 Ma (meaning after the MMCT). We have clarified this paragraph.

Section 2

Line 119: "location" more appropriate than "locale"?

> **Reply:** Changed per the reviewers suggestion. Thank you.

Line 120: please add water depth of Site 722.

> **Reply:** Done. This section was extended in the revised manuscript.

Lines 121-124: (a) Need to clarify relationship of Indian Ocean OMZ and Arabian Sea OMZ.

> **Reply:** Done. This section was expended and also has been better described in the introduction.

(b) Present-day water depth of Site 772 is 2028 m, which is well below the Arabian Sea OMZ (given as between 200 and 1000 m on Line 123). Please check references on the OMZ vertical extent at site location.

> **Reply:** Yes, this is correct. We are sorry it was not made clear we are fully aware of the fact the Site 722 lies below the OMZ in generally well oxygenated deep waters.

Line 125: according to ODP/IODP convention, site needs to be capitalized only when referring to a specific site (e.g., Site 722).

> **Reply:** We are aware, and apologize for not being consistent with formatting through the manuscript. We again went through the MS the make sure all issues with Site vs site and erroneous reference to Site 722B (see comment on Line 127 below) are not correct.

Line 127: should be Hole 722B (one of the holes drilled at Site 722) and not Site 722B.

> **Reply:** See above

Line 128: I guess you refer to Bialik et al., 2020a, when you mentioned "data used in this study". This is a bit confusing, so please give reference here.

> **Reply:** Done. We have extended this section to alleviate any potential imprecisions regarding this crucial detail.

Section 3

Line 147: Fig. 2 does not really show how the correction factor was derived and applied. Relevant information is required in Methods.

> **Reply:** This was revised to the full application of MARs according to the request of rev#1. The section was also extended to better detail all performed calculations

Line 173: replace "less than" by "fewer than".

> **Reply:** Thank you, other changes to this text changed the wording.

Line 190: data usually plural.

**Reply:** True, thank you for pointing out this error.

**Section 4**

Lines 191-193: verb appears to be missing.

    **Reply:** We added the missing verb ‚calculated'

Lines 197-199: verb tense should be consistent (either past or present).

    **Reply:** Thank you, we have fixed this error

Line 237: please provide brief information on preservation, as you did for calcareous nannofossils.

    **Reply:** Done, thank you for pointing out this oversight.

**Section 5**

Line 317: should be "delineate".

    **Reply:** Thank you.

Lines 353, 376, 443, 451, 486: please use either "foraminifer" or "foraminifera" consistently throughout the ms.

    **Reply:** done

Line 384: verb should be plural (are).

    **Reply:** done

Line 387: AABW and NADW usually referred to as Antarctic Bottom Water and North Atlantic Deep Water (not Waters).

    **Reply:** Thank you, we have fix this error.

Line 388: Did Woodruff and Savin (1989) specifically referred to this time interval? Precise dating of NADW expansion and AAWB intensification in this part of the Late Miocene remains controversial.

    **Reply:** To quote from Woodruff and Savin (1989): „*At the sametime, NADW beganto formwith greatenoughintensityto influenceSouthAtlanticbenthicforaminifera. At approximately11 Ma, NADW formation became more intense, and AABW in a form more similarto today'sbeganto flow into the Atlantic and Pacific. Thelatest Miocene thermohaline circulation was inmostrespectssimilarto that of the modern ocean*".

Line 412: why refer to Fig. 1 here?

    **Reply:** Since Site U1443 should be shown in figure one (which it isn't due to an error on our part). This was fixed in the revised version. We apologize for the confusion.

Line 461: should be "foraminifera"t to be consistent with remainder of text.

**Reply:** Done. Thank you

Lines 466-467: reference needed for this shift at 12 Ma.

**Reply:** Done

Line 515: there is a problem with the numbering of sections: duplication of headers (sections 5.1 and 5.2).

**Reply:** Thank you, we have fixed this formatting issue.

Lines 521-523: last part of sentence "paired with an expanded OMZ" appears disconnected from the first part of the sentence.

**Reply:** Indeed, we apologize. This issue was fixed.

Line 634: Taucher et al., 2022 is not an appropriate reference here.

**Reply:** Agreed, we removed the reference.

Line 711: these references are not appropriate here.

**Reply:** Agreed, they have been removed.

Line 731: ODP/IODP needs to be acknowledged for providing samples.

**Reply:** Done, we apologize for the oversight

Line 1234: Kuhnt et al. 2015 misspelt.

**Reply:** Thank you

Figures

Fig. 3: please specify calibration applied for TEX86 SST data from Zhuang et al. (2017).

**Reply:** Done.

Fig. 5: (a) Not really clear what is meant by "(note the abundance scaling of N*109/g)", which also appears in the plot of % diatom frustules. (b) The label "Planktonic foraminifera" should be indicated on the last plot. One has to read the caption to find out what is plotted there. (c) please provide reference(s) in figure caption for previously published data, where appropriate.

**Reply:** Revised as requested

References:

Holbourn, A.E., Kuhnt, W., Lyle, M., Schneider, L., Romero, O., and Andersen. N., 2014. Middle Miocene climate cooling linked to intensification of eastern equatorial Pacific upwelling. Geology, 42, 19–22, doi:10.1130/G34890.1.

Holbourn, A.E., Kuhnt, W., Kochhann, K.G.D., Andersen. N., and Meier, K.J.S., 2015. Global perturbation of the carbon cycle at the onset of the Miocene climatic optimum. Geology, 43, 123–126, doi:10.1130/G36317.1.